# OmniTalker: One-shot Real-time Text-Driven Talking Audio-Video Generation With Multimodal Style Mimicking

**Zhongjian Wang**[*]   **Peng Zhang**[*]   **Jinwei Qi**[*]

Guangyuan Wang     Sheng Xu     Bang Zhang

Tongyi Lab, Alibaba Group

Project Page: https://humanaigc.github.io/omnitalker

## Abstract

Although significant progress has been made in audio-driven talking head generation, text-driven methods remain underexplored. In this work, we present OmniTalker, a unified framework that jointly generates synchronized talking audio-video content from input text while emulating the target identity's speaking and facial movement styles, including speech characteristics, head motion, and facial dynamics. Our framework adopts a dual-branch diffusion transformer (DiT) architecture, with one branch dedicated to audio generation and the other to video synthesis. At the shallow layers, cross-modal fusion modules are introduced to integrate information between the two modalities. In deeper layers, each modality is processed independently, with the generated audio decoded by a vocoder and the video rendered using a GAN-based high-quality visual renderer. Leveraging DiT's in-context learning capability through a masked-infilling strategy, our model can simultaneously capture both audio and visual styles without requiring explicit style extraction modules. Thanks to the efficiency of the DiT backbone and the optimized visual renderer, OmniTalker achieves real-time inference at 25 FPS. To the best of our knowledge, OmniTalker is the first one-shot framework capable of jointly modeling speech and facial styles in real time. Extensive experiments demonstrate its superiority over existing methods in terms of generation quality, particularly in preserving style consistency and ensuring precise audio-video synchronization, all while maintaining efficient inference.

## 1   Introduction

Recent advancements in talking head generation (THG) have been predominantly driven by breakthroughs in generative architectures. Although current research predominantly focuses on audio-driven THG systems[37, 57, 45, 62, 16, 60], emerging applications in conversational AI and human-computer interaction increasingly demand text-driven solutions - particularly given the paradigm shift enabled by large language models (LLMs). Despite this technological imperative, text-driven THG methodologies remain comparatively underdeveloped relative to their audio-driven counterparts.

Existing text-driven approaches[61, 59, 53, 43, 7, 24] typically employ a cascaded architecture combining text-to-speech (TTS) systems with audio-driven THG models. This conventional paradigm suffers from three fundamental limitations: (1) computational redundancy through duplicated feature processing, (2) error propagation across disjoint subsystems, and (3) audio-visual style discrepancies. Recent attempts to address these issues, such as UniFlug[35] that utilizes TTS latent features for facial keypoint generation, remain constrained by unidirectional audio-to-visual information flow.

---

[1]Equal Contributions

39th Conference on Neural Information Processing Systems (NeurIPS 2025).

While methods like [3, 34] demonstrate end-to-end generation capabilities, they require training or finetuning for every identity, severely limiting practical deployment..

The synthesis of expressive talking heads further necessitates precise modeling of multimodal speaking styles that simultaneously capture vocal characteristics, head motion patterns, and facial dynamics. Current solutions predominantly address style modeling in audio-driven contexts through either categorical emotion labels[46, 19, 41] or video-based motion aggregation[33, 50]. However, these approaches fail to holistically represent the complex interplay between acoustic and visual style components that characterize natural human communication.

To overcome these challenges, we present OmniTalker, an end-to-end framework for one-shot text-driven talking head generation with unified audio-visual style transfer. Our architecture integrates three key innovations: (1) A multimodal diffusion transformer (DiT) backbone enabling bidirectional cross-modal attention across audio and visual streams; (2) Dual transformer decoders for modality-specific refinement while preserving cross-contextual information; and (3) A masked infilling strategy that leverages DiT's in-context learning capabilities for joint audio-visual style transfer without dedicated style extraction modules. The complete system is trained on a large-scale multimodal dataset.

Our principal contributions can be summarized as follows:

- We propose OmniTalker, the first end-to-end unified framework that generates synchronized, high-quality audio-visual talking heads directly from text input in a one-shot learning paradigm.

- OmniTalker represents the first comprehensive solution for fully replicating an individual's speaking style, including the dynamic triad of vocal characteristics, head motion patterns, and facial expression dynamics.

- The proposed architecture achieves real-time inference on a single NVIDIA RTX 4090 GPU. It surpasses existing approaches in generation quality, with particular improvements in style preservation and audio-visual synchronization precision.

## 2 Related Work

We present the most relevant lines of work here, but since talking head research is vast, a detailed comparison with prior methods is included in the Appendix A.

### 2.1 Audio-driven Talking Head Generation

Early audio-driven THG works[37, 16] primarily focuses on lip-synchronization by modifying mouth regions in target videos based on audio input. Recent advancements have extended this paradigm to generate talking head videos from single reference images [45, 33, 14, 57]. Most state-of-the-art methods adopt a two-stage framework: (1) mapping audio signals to intermediate motion representations (e.g., 3DMM coefficients [62, 63], facial landmarks [64, 27, 58], or learnable latent codes [57, 23]), followed by (2) video synthesis conditioned on the predicted motion. While these methods have achieved remarkable progress, their reliance on audio input constrains practical applications in scenarios where only textual input is available.

Our work inherits the two-stage generation paradigm for its effective decoupling of identity, head pose, and facial expressions. However, we propose a novel end-to-end architecture that predicts head pose and facial expressions directly from textual input rather than audio signals.

### 2.2 Style-Controlled Talking Head Generation

Existing approaches typically adopt simplified formulations: some works [46, 19, 41] represent styles as discrete emotion categories, while others [28, 22] employ reference videos for frame-level expression control - an approach that fails to capture the temporal dynamics of facial expressions. StyleTalk [33] introduces a more sophisticated solution by extracting spatiotemporal style codes from reference videos, though this method primarily focuses on expression styles and neglects head pose variations.

Current literature predominantly reduces speaking style to facial dynamics, overlooking two critical aspects: (1) the inherent correlation between vocal and facial styles, and (2) the semantic dependency between speaking style and linguistic content. This limitation motivates our work to develop a more comprehensive style modeling framework.

## 2.3 Text-driven Talking Head Generation

Text-driven talking head generation systems aim to jointly synthesize speech audio and corresponding facial animations from textual input. The predominant approach employs a cascaded pipeline combining a text-to-speech (TTS) module with an audio-driven talking head generator [61, 59, 53, 43, 35, 7, 24]. Cascaded systems are inherently limited by three key issues: (1) computational redundancy through duplicated feature processing, (2) error propagation across disjoint subsystems, and (3) mismatches between audio and visual stylistic characteristics.

Recent efforts have explored parallel generation architectures to address these issues [9, 34, 35, 21, 3]. Among existing solutions, TTSF [21], NEUTART, and AV-Flow [3] share similarities with our approach in employing cross-modal conditioning. However, TTSF predicts sound directly from input images, which, while reducing dependency, fails to replicate the specified identity. Both AV-Flow and NEUTART are person-specific methods, requiring training for individual subjects.

## 3 Method

We aim to perform joint audio-visual generation using a compact neural architecture that ensures alignment between audio and visual modalities while preserving the speaking style (both vocal and facial) from a reference video. Inspired by the in-context reference commonly used in LLM[2] and TTS[4], as well as the dual-single stream hybrid diffusion transformer(DiT) structure utilized in text-to-image synthesis[13, 25], we propose the network architecture illustrated in Figure 1. The architecture integrates three core components: (1) Multi-Modal feature extractors to capture reference dynamics, (2) a dual-branch DiT network for parallel audio-visual synthesis, and (3) an Audio-Visual fusion module ensuring tight synchronization. Unlike previous approaches, we aim at the domain of multimodal generation.

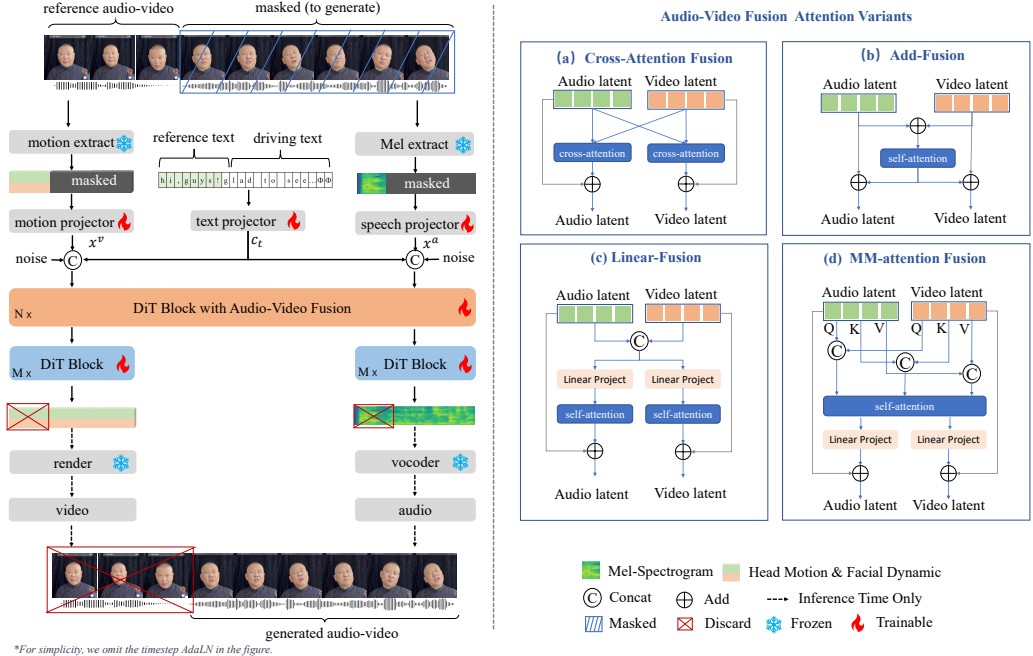

Figure 1: The architecture of OmniTalker. A dual-branch DiT framework with cross-modal fusion for joint audio-video generation from text(left). Four variants of Audio-Video fusion modules are explored(right).

## 3.1 Multi-Modal Feature Alignment

Our model takes two inputs: the driven text $T_d$ and a style reference video, to produce a realistic talking face video. We first separate the audio and video streams from the reference video, denoting them as $A_r$ and $V_r$, respectively. ASR models are adopted to convert $A_r$ into transcript text $T_r$ as reference text.

**Audio Feature** A mel-spectrogram $M_r \in \mathbb{R}^{F \times N_a}$ is extracted from $A_r$, where $F$ is the mel dimension and $N_a$ is the length of the reference sequence(at 94 fps by default). A MLP-based module integrates the mel-spectrogram, to obtain the audio embeddings $x^a$.

**Visual Feature** We extract visual codes $C_r \in \mathbb{R}^{61 \times N_v}$ for each video clip, which consists of facial expression blendshapes $\alpha_{exp} \in \mathbb{R}^{51}$, head pose $[R;t] \in \mathbb{R}^6$ and eye movement coefficients $\alpha_{eye} \in \mathbb{R}^4$ for each frame. Blendshape is adopted for its more explicit semantic interpretations compared to the commonly used 3DMM and FaceVerse[49] coefficients. We firstly detect facial landmarks $V_{2d}$ from each frame and optimizes $C_r = [\alpha_{exp}, \alpha_{eye}, R, t]$ by minimizing distances between the projected keypoints and the detected landmarks as follows:

$$\mathcal{L}_{lmk} = \|P_r \times T \times \begin{bmatrix} \bar{S} + A_{exp}\alpha_{exp} + A_{eye}\alpha_{eye} \\ 1 \end{bmatrix} - V_{2d}\|^2 \tag{1}$$

where $P_r$ is the orthographic projection matrix, $T \in \mathbb{R}^{3 \times 4}$ is the similarity transformation matrix constructed by $[R;t]$, $\bar{S}$ is the mean 3D shape, $A_{exp}$ is the expression base and $A_{exp}$ is the eye movement base. $N_v$ denotes the number of frames in the video sequence, which is 30 fps by default. To align $N_v$ with $N_a$, we upsample $C_r$ to 94 fps through interpolation. $C_r$ is then projected to the visual embeddings $x^v$ by a MLP-based embedding module.

**Input Text** Both the driving text $T_d$ and the reference text $T_r$ are converted into a pinyin sequence (for Chinese) or character/letter sequences (for Latin-based languages)[4]. The two character sequences are concatenated and padded with filler tokens $\Phi$ of the same length as $x^a$. Let $S_T$ represent the final sequence. The text embedding module is built upon ConvNeXt-V2[55], which is a common approach in the field of TTS, due to its strong temporal modeling capability. The projected text embeddings, denoted $c_t$ serves as condition for both audio and visual branches.

## 3.2 Multi-Modal Feature Fusion

Figure 1 illustrates our network, which consists of multiple DiT blocks designed to handle both audio and visual data streams. The network is structured to facilitate the cross-modal fusion between audio and visual features, enabling the generation of coherent and synchronized outputs.

**Text Conditioning** As detailed in Section 3.1, the model architecture processes three modalities: text (as character sequences), audio, and visual features. For the text modality, we apply an absolute sinusoidal position embedding before feeding it into the ConvNeXt blocks. This design allows the text representation to develop a moderate level of modality-specific modeling capacity through the ConvNeXt blocks, enabling better alignment with other modalities before cross-modal fusion and in-context learning with other inputs. Diverging from MMAudio's [6] approach that maintains a separate text branch, our architecture directly concatenates text embeddings with the audio and visual modalities. This decision is motivated by our observation that the pure MMDiT structure may be overly flexible for tasks requiring high-fidelity generation aligned with explicit prompt guidance, e.g. TTS and THG. To preserve spatial coherence in the multimodal input, we augment the concatenated sequences with convolutional position embeddings.

**Audio-Visual Fusion Module**: This module is responsible for the integration of audio and visual features. It employs a dual-branch architecture, where one branch processes visual information, and the other processes audio. As for feature fusion mechanism, common strategies include (a) Cross-Attention, (b) Element-wise Addition, (c) Linear Fusion[3], (d) Multi-modal joint attention(MM-Attention)[13, 25, 6], as shown in Figure 1. After empirical validation, we adopted MM-Attention which demonstrated superior performance. We will provide a detailed comparison of the effects of different fusion methods in Section 4.4. In MM-Attention, the Query($Q$), Key($K$), and Value($V$) matrices are derived from both modalities in joint attention, and RoPE[44] is applied. MM-Attention allows the network to dynamically weight the importance of audio and visual features, ensuring that the generated video is temporally aligned with the input audio.

**Single-Modality DiT Blocks**: Following the Audio-Visual Fusion Module, the network employs several single-modality DiT blocks. These blocks operate on the fused multimodal features but are designed to refine the generation process by focusing on individual modalities (audio or visual) after the initial cross-modal fusion. This two-stage approach, first fusing multimodal information and then refining each modality separately, allows the network to maintain the coherence of the generated content while ensuring high-quality output for each modality.

### 3.3 In-Context Stylized Audio-Visual Generation

We propose an audio-visual sequence infilling framework that predicts target segments by leveraging contextual information from surrounding segments and full text transcriptions (including both reference text and driving text). During training, we implement a masked reconstruction strategy where multiple random segments in the audio-visual sequence are occluded. The model is optimized through audio-visual reconstruction loss computed on these masked regions. This in-context reference mechanism enables the network to implicitly learn stylistic features from reference videos without requiring complex additional style extraction modules. For inference, our approach employs a three-component input structure: (1) a reference audio-visual pair serving as the talking style prompt, (2) arbitrary driving text as the conditional input, and (3) a noisy latent placeholder representing the target audio-visual sequence to be predicted. This architecture allows the model to synthesize audio-visual synchronized sequences that inherit the stylistic characteristics defined in the reference prompt while maintaining alignment with the provided textual condition, as illustrated in Figure 1.

**Audio-Visual Generation via Flow Matching** We employ flow matching[30] to train our DiT-based model, leveraging its superior advantages such as improved training and inference efficiency, simplified optimization, and faster convergence compared to traditional methods. Let $x \in \mathbb{R}^d$ donate the data points(mel-spectrogram $M_r$ and visual codes $C_r$ in our case), sampled from the ground truth data distribution $q(x)$, and $p : [0,1] \times \mathbb{R}^d \to \mathbb{R} > 0$ donate the probability density path, which is a time dependent probability density function. Our model with parameters $\theta$ predict a vector field $v_t$, which can be used to construct a time-dependent flow, $\phi : [0,1] \times \mathbb{R}^d \to \mathbb{R}^d$ via the ordinary differential equation (ODE). Follow the Optimal Transport (OT) formulation from [30], we can train our model with conditions $C = [M_r, C_r, S_T]$ by optimizing the conditional flow matching objective:

$$\mathcal{L}_{CFM}(\theta) = \mathbb{E}_{t,q(x_1),p(x_0)} \left\| v_t((1-t)x_0 + tx_1, C; \theta) - (x_1 - x_0) \right\|^2 \tag{2}$$

We provide detailed preliminaries of flow matching in Appendix B.1. In the training stage, we pass in the model the noisy audio-visual embeddings $(1-t)x_0 + tx_1$ and the masked embeddings $(1-m) \odot x_1$, where $x_1 = [M_r; C_r]$, $x_0$ denotes sampled Gaussian noise and $m \in \{0,1\}^{(F+61) \times N_a}$ represents a binary temporal mask. The model is trained to reconstruct $m \odot x_1$ with $(1-m) \odot x_1$ and $S_T$. We apply the conditional flow matching loss function formulated in Equation (2) for both the audio and visual branches. The overall objective is:

$$\mathcal{L} = \lambda_m \mathcal{L}_{CFM}^m + \lambda_h \mathcal{L}_{CFM}^h + \lambda_f \mathcal{L}_{CFM}^f + \lambda_e \mathcal{L}_{CFM}^e \tag{3}$$

where $m$ is for mel-spectrogram, $h$ is for head pose, $f$ is for facial expressions and $e$ is for eye movements correspondingly. We use L1 loss instead of L2 as we found that an L1 loss leads to more realistic results. During inference, we could push the data from the Gaussian distribution to the target distribution by any ODE solver.

**Classifier-Free Guidance** To further improve generation quality and enhance style replication, we integrate classifier-free guidance (CFG)[20] into both branches. We randomly drop each of the input conditions $C = [M_r, C_r, S_T]$ in the training stage, while during inference, we apply multi-conditions CFG to construct the $v_t^{CFG}$:

$$v_t^{CFG} = (1 + \sum_{c \in C} \alpha_c) \cdot v_t(x_0, c) - \sum_{c \in C} \alpha_c \cdot v_t(x_0, 0) \tag{4}$$

, where $\alpha_c$ is the guidance strength for condition $c$. Please refer to Section 4.1 for the detailed parameter settings mentioned above.

### 3.4 Audio And Video Decoders

The synthesized mel-spectrogram is decoded to speech via Vocos[42], a high-fidelity vocoder selected for its real-time efficiency. Alternative vocoders (e.g., BigVGAN[26]) can be flexibly integrated.

We design a warp-based GAN model as the visual render(similar to [57, 11], which randomly selects a frame from the reference video as the identity reference and generates video frames based on the visual codes $C'_r$(including head pose, blendshapes and eye movements) predicted by the DiT network. It produces $512 \times 512$ resolution frames at 50 fps, ensuring real-time performance. While the visual rendering pipeline is beyond the scope of this work, it is designed to be both computationally efficient and highly flexible. We refer readers to the Appendix B.2 for more details.

## 4 Experiments

### 4.1 Experimental Setup

**Implementation Details.** Our model includes 22 audio-visual fusion blocks with two parallel branches (4 single-modality DiT blocks each), 512-dim embeddings (audio/visual via linear layers, text via 4 ConvNeXt V2 blocks), totaling 0.8B parameters. The model was trained on 8 NVIDIA A100 GPUs using a batch size of 12,800 frames per GPU for 750,000 iterations with a learning rate of $1e-4$. In Equation (3), we use $\lambda_m = 0.1$, $\lambda_f = 3.0$, $\lambda_h = 0.5$ and $\lambda_e = 0.5$. The audio features are represented as 100-D$(F = 100)$ log-mel filterbank coefficients extracted with 24kHz sampling rate and hop length 256, yielding an audio sequence with approximately 94 fps. In contrast, visual codes are captured at 30 fps and upsampled to 94 fps to match the audio frame rate. During inference, the duration of reference audio-visual content is restricted to 1–10 seconds, with any excess truncated. The CFG parameters are set to $\alpha_{M_r} = 2.0$, $\alpha_{C_r} = 2.5$ and $\alpha_{S_T} = 2.0$, while 16 sampling steps are used. We provide full details in Appendix C.

**Datasets.** We evaluated our method on both audio and video generation tasks. For audio, we employed the SEED[1] dataset, while for video generation, 500 clips were randomly selected from VoxCeleb2[8] as the test set. To further validate the robustness of our approach in practical applications, 100 video clips from Chinese real-world scenarios were selected as ***Custom*** dataset for supplementary testing of audio and video quality. We pre-trained our model on a collection of large-scale open-source talking-head datasets and subsequently developed a high-quality dataset(totaling 690 hours) to fine-tune the model's performance. We direct the authors to Appendix C.1 for more details.

**Compared Baselines.** Since there is currently no method capable of jointly generating audio and video simultaneously, we conducted comparative analyses on text-to-speech (TTS) methods and audio-driven talking head approaches separately. For the TTS module comparison, we selected three representative methods: MaskGCT[54], F5TTS[4], and CosyVoice[12]. Since SEED lacks visual information, we set $\alpha_{C_r} = 0.0$(w/o visual condition) during testing. On the ***Custom*** dataset, we separately evaluated the scenarios of $\alpha_{C_r} = 0.0$ and $\alpha_{C_r} = 2.5$(w. visual condition). In the evaluation of Talking Head generation modules, we established a multimodal comparative framework

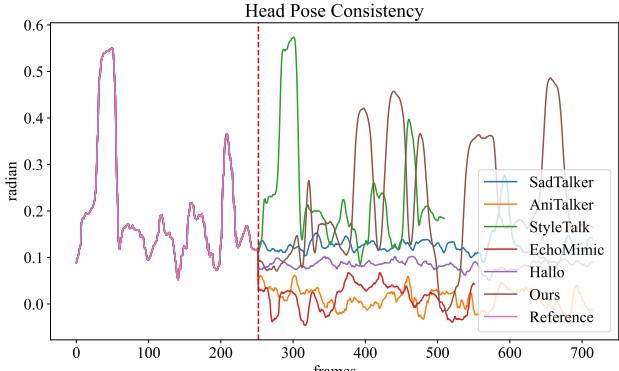

Figure 2: We visualize the values of head poses over time for the reference video and videos generated by different methods. To better showcase the differences, we focus on displaying the Yaw angle. The red dashed line separates the values of the reference video on the left from those of the generated results on the right. *StyleTalk\*\* can only mimic the expression style and simply copy the reference head pose*.

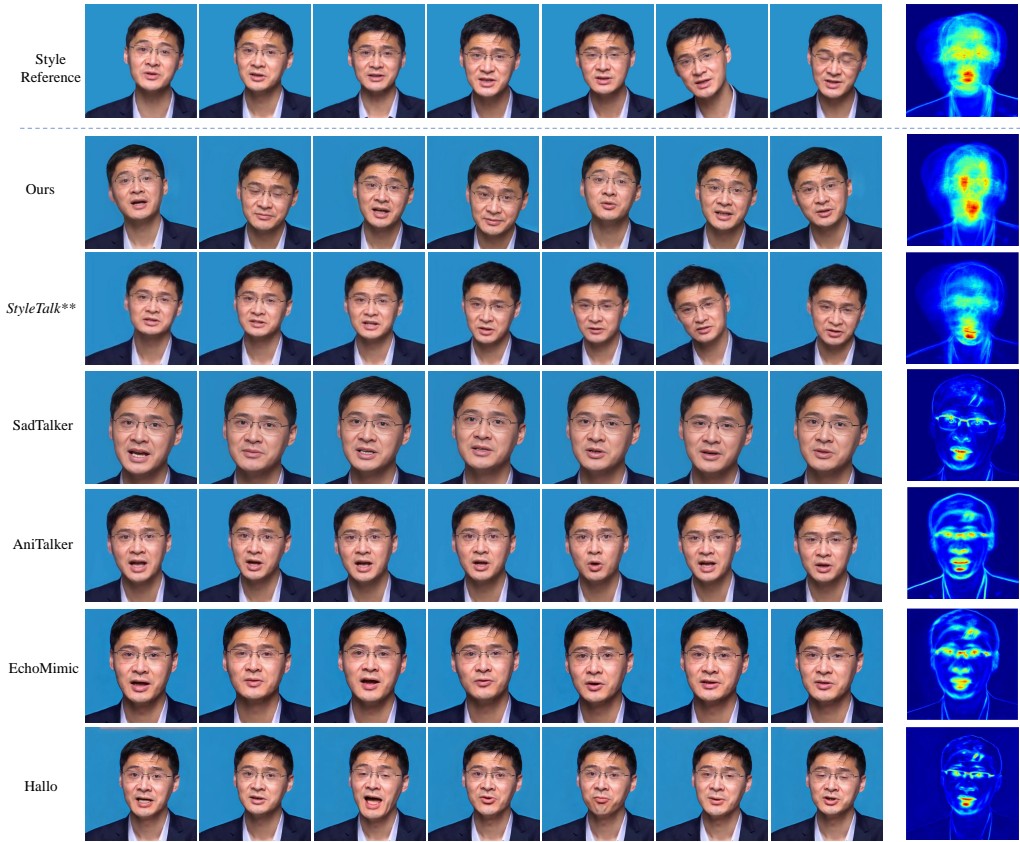

Figure 3: **Qualitative Comparision.** Given a video for identity and style reference, we visualize generated frames of all compared methods. We also visualize the motion heatmaps of generated videos on the right side. For the audio-driven approach, we use the audio output from our own method as the input. *StyleTalk\*\* directly transfers the reference pose sequence.*

encompassing: (1) two GAN[15] based methods (SadTalker[62] and AniTalker[31]); (2) two state-of-the-art diffusion model-based approaches (EchoMimic[5] and Hallo[56]); and (3) StyleTalk[33] – an audio-driven THG method with style-preservation capabilities. It is important to note that all talking head models in our experiments employed audio signals generated by our proposed method as the driving input, ensuring fairness and comparability across evaluations. All methods were evaluated at $512 \times 512$ with a single NVIDIA RTX 4090 GPU, whereas Sadtalker and AniTalker employ a super-resolution model by default.

**Evaluation Metrics.** We evaluate audio quality using the word error rate (WER), measured with the Whisper-Large-v3[38] model. Additionally, we employ speaker similarity (SIM-A) to assess the performance of zero-shot timbre preservation. The evaluation metrics utilized in the portrait image animation approach include Fréchet Video Distance (FVD)[48], Synchronization-C (Sync-C)[40]. Specifically, FVD measures the similarity between generated videos and real data, with lower values indicating better performance, and thus more realistic outputs. Sync-C evaluates the lip synchronization of generated videos in terms of content and dynamics, with higher Sync-C scores denoting better alignment with audio. We also compute the cosine similarity(CSIM) between facial identity features of the reference image and generated frames. Following EMO[47, 56], we assess the preservation of speaking style by calculating the Fréchet distance between motion coefficients extracted from the reference and generated videos. To separately evaluate expression and pose fidelity, we introduce E-FID (Expression FID) and P-FID (Pose FID). Specifically, E-FID integrates facial blendshapes $\alpha_{exp}$ and eye movements $\alpha_{eye}$, while P-FID quantifies head pose consistency through $[R; t]$ parameters.

## 4.2 Quantitative Evaluation

Table 1: Quantitative comparison with TTS methods.

| Method | SEED | | | | *Custom* | |
|---|---|---|---|---|---|---|
| | ZH | | EN | | ZH | |
| | WER(%)↓ | SIM-A↑ | WER(%)↓ | SIM-A↑ | WER(%)↓ | SIM-A↑ |
| CosyVoice[12] | 3.10 | 0.723 | 4.29 | 0.609 | 3.52 | 0.725 |
| MaskGCT[54] | 2.27 | **0.774** | 2.62 | **0.714** | 2.83 | 0.769 |
| F5-TTS[4] | 1.56 | 0.741 | **1.83** | 0.647 | 1.82 | 0.762 |
| **Ours**($\alpha_{C_r} = 0$) | **1.55** | 0.766 | 1.85 | 0.685 | 1.61 | **0.771** |
| **Ours**($\alpha_{C_r} = 2.5$) | - | - | - | - | **1.58** | 0.762 |

Table 2: **Quantitative comparison** with SOTA audio-driven talking head generation methods. We highlight the **best**, second best.

| Method | VoxCeleb2 | | | | | *Custom* | | | | | Inference |
|---|---|---|---|---|---|---|---|---|---|---|---|
| | FVD↓ | CSIM↑ | Sync-C↑ | E-FID↓ | P-FID↓ | FVD↓ | CSIM↑ | Sync-C↑ | E-FID↓ | P-FID↓ | FPS↑ |
| SadTalker[62] | 243.92 | 0.828 | 5.34 | 0.67 | 3.37 | 595.00 | 0.854 | 3.61 | 0.45 | 9.23 | 25 |
| AniTalker[31] | 181.58 | 0.842 | 4.82 | 0.94 | 1.08 | 329.70 | 0.858 | 4.02 | 0.89 | 3.64 | 18 |
| StyleTalk[33] | 144.40 | 0.691 | 5.52 | 0.71 | 0.31 | 208.51 | 0.791 | 4.17 | 0.51 | 0.40 | 21 |
| EchoMimic[5] | 191.77 | 0.883 | 5.88 | 0.60 | 2.74 | 367.74 | **0.908** | 4.74 | 0.36 | 8.26 | 0.78 |
| Hallo[56] | 156.33 | 0.879 | **6.53** | 0.59 | 1.03 | 244.40 | 0.887 | **5.52** | 0.35 | 1.06 | 0.65 |
| **Ours** | **102.54** | **0.890** | 6.37 | **0.02** | **0.04** | **176.32** | 0.905 | 5.48 | **0.08** | **0.12** | 25 |

**Audio Results.** In quantitative analysis of audio generation as demonstrated in Table 1, our approach exhibits significant superiority over TTS baseline models. WER is notably reduced on both SEED-ZH and *Custom* test sets, while showing only marginal underperformance compared to F5TTS on SEED-EN, demonstrating that the generated audio strictly adheres to the input text. Additionally, our method ranks second in SIM-A metrics, further validating the preservation capability of identity features under zero-shot conditions. Notably, in the *Custom* dataset, incorporating visual conditions achieves lower WER values, indicating that introducing visual supervision effectively enhances perceptual audio quality. We attribute this improvement to the effectiveness of multi-task learning, as audio signals and facial motion patterns represent highly correlated modalities.

**Video Results.** Table 2 further demonstrates the superior performance of our method in visual generation. On both the VoxCeleb2 and *Custom* datasets, our approach achieves SOTA performance on 5 core metrics, with the lowest FVD and highest CSIM scores. These metrics confirm our model's significant advantage in generating high-quality videos while preserving identity consistency. Notably, our method achieves orders-of-magnitude improvements on the E-FID and P-FID metrics compared to existing approaches, highlighting its exceptional capability in preserving facial motion patterns and head pose characteristics. This enables effective inheritance of the reference person's speaking style, leading to high-fidelity expression cloning with remarkable authenticity. While our method obtains suboptimal results on the Sync-C, slightly underperforming the diffusion-based method, empirical analysis reveals that these metrics favor front-facing video generation. In contrast, baseline methods tend to produce front-facing videos at the expense of directional information from the reference images. Comparison videos are provided in the supplementary materials for further insight.

**User Study.** We performed an MOS evaluation (1–5) based on [60] with 50 volunteers to assess perceptual quality. Participants rated videos on six aspects: speech similarity, rhythm, visual quality, speaking style, lip-sync, and pose-sync, details in Appendix C.4. The results are shown in Table 3. Our method achieved the best performance in speech similarity, visual quality, speaking style, and pose synchronization, with significant improvements in style preservation and pose generation over competing methods, highlighting our state-of-the-art capabilities in maintaining natural speaking styles and generating synchronized facial movements.

**Inference Efficiency.** Our method achieves high-quality output with real-time inference capabilities through the innovative integration of flow matching technology and a relatively compact model architecture (with only 0.8 billion parameters). As shown in Table 2, our method outperforms others in both inference speed and output quality. **It is noteworthy** that the fps number of baseline methods do not incorporate the inference time of the preceding TTS process. In other words, none of these methods can truly achieve real-time performance in practice.

Table 3: MOS score of different methods by user study.

|  | CosyVoice | MaskGCT | F5TTS | SadTalker | AniTalker | StyleTalk | EchoMimic | Hallo | Ours |
|---|---|---|---|---|---|---|---|---|---|
| Speech Similarity | 3.56 | 4.32 | 4.40 | - | - | - | - | - | **4.66** |
| Speech Rhythm | **4.24** | 3.88 | 3.90 | - | - | - | - | - | 4.02 |
| Visual Quality | - | - | - | 3.86 | 3.90 | 3.32 | 3.96 | 4..00 | **4.12** |
| Speak Style | - | - | - | 2.10 | 2.56 | 3.98 | 3.88 | 3.94 | **4.57** |
| Lip-sync | - | - | - | 4.00 | 3.90 | 4.00 | 3.82 | **4.30** | 4.20 |
| Pose-sync | - | - | - | 1.22 | 1.60 | 1.98 | 3.00 | 3.52 | **4.44** |

Table 4: Ablation study of Fusion Strategies.

| Fusion | WER(%)↓ | SIM-A↑ | Sync-C↑ | E-FID↓ | P-FID↓ |
|---|---|---|---|---|---|
| Add | 2.89 | 0.582 | 4.92 | 0.58 | 3.20 |
| Linear | 2.32 | 0.645 | 5.08 | 0.21 | 1.54 |
| Cross-Attention | 1.72 | 0.698 | 5.35 | 0.12 | 0.18 |
| MM-Attention | **1.58** | **0.762** | **5.48** | **0.08** | **0.12** |

## 4.3 Qualitative Evaluation

**Facial Motion Styles.** To more intuitively demonstrate the superior capability of our method in modeling facial motion styles, we compared the cumulative heatmaps of head movements in speaking videos generated by our method and other existing approaches. As shown in Figure 3, by comparing the generated results with reference videos, it is evident that the heatmaps produced by our method align more closely with those of the real data. This result indicates that our method possesses a significant advantage in capturing and reproducing complex facial motion styles. Figure 2 further validates this conclusion from a temporal perspective. We selected the yaw angle (a parameter of head pose) as the tracking metric. The red line on the left represents the reference sequence, while the right side displays the sequences generated from various methods. The results show that the generated sequences of our method maintain a high consistency with the reference sequence in terms of amplitude and frequency of movement while preserving necessary differences, demonstrating effective inheritance of the head-pose style. In contrast, the head movements generated by other methods were generally less pronounced and lacked dynamic variation. It should be noted that the StyleTalk method directly transfers the reference pose sequence to achieve generation, which ensures complete consistency with the reference pose but fails to consider the semantic association between speech content and pose.

## 4.4 Ablation Study

**Fusion Strategies.** As shown in Table 4, the MM-Attention method achieved the best performance in all terms. This suggests that employing MM-Attention as the fusion strategy for audio and motion features may be the most effective approach. It better captures the correlations and importance distributions between the two modalities, thereby enhancing the overall performance. Although others can also achieve feature fusion, their performance generally lags behind that of MM-Attention. In particular, the Add method performed the worst in all evaluation metrics. This may be because simply adding or linearly combining the two features fails to adequately account for their complex interactions and the nuanced importance distributions inherent to each modality.

## 5 Conclusion

OmniTalker introduces a unified framework for text-driven talking head generation, achieving simultaneous audio-visual synthesis with in-context style replication. By integrating a dual-branch architecture with cross-modal attention mechanisms and in-contex style reference, the method bridges the gap between text input and multimodal output, ensuring audio-visual synchronization and stylistic consistency without reliance on cascaded pipelines. The combination of a lightweight network architecture and flow matching based training enables the model to deliver high-quality output with real-time inference capabilities. Extensive experiments demonstrate that our method surpasses existing approaches in generation quality, particularly excelling in style preservation and audio-video synchronization, while maintaining real-time prediction efficiency.

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

# A  Comparison with Prior Methods

Table 5: Capabilities of different methods

| Method | text driven | one-shot | voice clone | style replicate | real-time |
|---|---|---|---|---|---|
| Wav2Lip[37] | | ✓ | | | ✓ |
| VividTalk[45] | | ✓ | | | ✓ |
| SadTalker[62] | | ✓ | | | ✓ |
| StyleTalk[33] | | ✓ | | ✓ | |
| VASA-1[57] | | ✓ | | | ✓ |
| EchoMimic[5] | | ✓ | | | |
| Hallo[56] | | ✓ | | | |
| TTSF[21] | ✓ | ✓ | | | ✓ |
| AV-Flow[3] | ✓ | | ✓ | ✓ | ✓ |
| Ours | ✓ | ✓ | ✓ | ✓ | ✓ |

# B  Model Details

Our model includes 22 audio-visual fusion blocks with two parallel branches (4 single-modality DiT blocks each), 512-dim embeddings (audio/visual via linear layers, text via 4 ConvNeXt V2 blocks), totaling 0.8B parameters. We provide detailed configurations in Table 6

Table 6: Model Configuration

| Module | Hyper Parameter | Value |
|---|---|---|
| TextEmbedding | ConvNeXt-V2 blocks | 4 |
| | Embedding dimension | 512 |
| | FFN dimension | 1024 |
| AudioEmbedding | Linear Layer | 1 |
| | Embedding dimension | 1024 |
| VisualEmbedding | Linear Layer | 1 |
| | Embedding dimension | 1024 |
| Audio-visual Fusion | Transformer blocks | 22 |
| | Attention heads | 16 |
| | Embedding dimension | 1024 |
| | FFN dimension | 2048 |
| Audio DiT branch | Transformer blocks | 4 |
| | AdaLayerNorm | 1 |
| | Linear Layer | 1 |
| | Attention heads | 16 |
| | Embedding dimension | 1024 |
| | FFN dimension | 2048 |
| Visual DiT branch | Transformer blocks | 4 |
| | AdaLayerNorm | 1 |
| | Linear Layer | 1 |
| | Attention heads | 16 |
| | Embedding dimension | 1024 |
| | FFN dimension | 2048 |

## B.1  Preliminaries on Flow Matching

Flow matching, evolved from Continuous Normalizing Flows (CNFs)[30], aims to learn a model that transforms a simple distribution $p_0$ into a more complicated one $p_1$. This objective aligns closely with the fundamental goal of diffusion models. The learning process is achieved by minimizing the difference between the flow of the data and the flow predicted by the model, with a simple objective:

$$\mathcal{L}_{FM}(\theta) = \mathbb{E}_{t,p_t(x)} \|v_t(x) - u_t(x)\|^2 \tag{5}$$

where $x$ denotes data points, $p_t(x)$ represents a time dependent probability density path, and $u_t(x) = \frac{dx_t}{dt}$ denotes the unknown time-dependent vector field governing the trajectory of the data distribution from from $p_0$ to $p_1$.

Specifically, given a specific sample $x_1$ from some unknown data distribution $q(x_1)$, $p_t(x|x_1)$ refers to the conditional probability path. The marginal probability path can be obtained by taking the expectation of the conditional probability path over all samples in the data distribution:

$$p_t(x) = \int p_t(x|x_1)q(x_1)dx_1 = \mathbb{E}_{q(x_1)}(p_t(x|x_1)) \tag{6}$$

Assuming $p_t(x|x_1)$ is derived from the conditional vector field $u_t(x|x_t)$, it follows that $u_t(x)$ and $u_t(x|x_t)$ satisfy the following relationship:

$$u_t(x) = \int u_t(x|x_1)\frac{p_t(x|x_1)q(x_1)}{p_t(x)}dx_1 = \mathbb{E}_{p(x_1|x)}(u_t(x|x_1)) \tag{7}$$

Therefore, the original $\mathcal{L}_{FM}(\theta)$ can be reformulated as:

$$\mathcal{L}_{CFM}(\theta) = \mathbb{E}_{t,q(x_1),p_t(x|x_1)} \|v_t(x;\theta) - u_t(x|x_1)\|^2 \tag{8}$$

We consider the case of Gaussian distributions and adopt a simple yet effective approach: the optimal transport (OT) path. Given the target data point $x_1$ and the current data point $x_t$, the most efficient way is to go with a straight line. The conditional vector field is then defined as $u_t(x|x_1) = (x_1-x)/(1-t)$. In this case, the CFM loss takes the form:

$$\mathcal{L}_{CFM}(\theta) = \mathbb{E}_{t,q(x_1),p(x_0)} \|v_t((1-t)x_0 + tx_1;\theta) - (x_1 - x_0)\|^2 \tag{9}$$

as described in Section 3.3

## B.2 Render

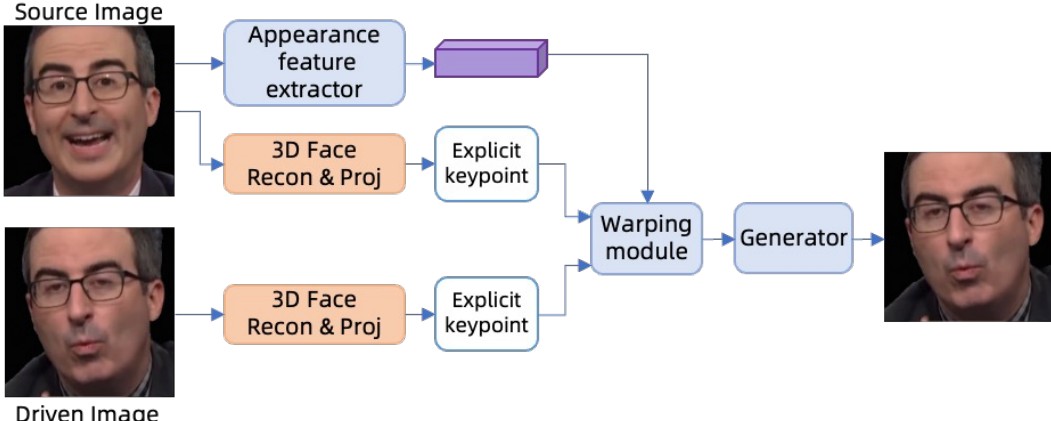

Figure 4: Overview of visual renderer.

To balance the quality of generated videos and model inference performance, we employ a warping-based GAN framework inspired by existing works [51, 18]. The general framework consists of four key components as shown in Figure 4: 1) *Appearance feature extraction* to get visual features from source image; 2) *Motion representation extraction* that extracts explicit facial keypoint to represent face movement; 3) *Warping field estimation* to calculate the transformation from source to target; and 4) *Generator* to synthesize final images from warped appearance features. Specifically, face motion is represented by 3D keypoints projected from a 3DMM head template [49], and 3D keypoints on key face regions are projected from the head template, including eyes, mouth, eyebrows and face contours, driven by 3DMM coefficients, which are capable of representing a diverse range of facial expressions.

We follow LivePortrait [18] using perceptual loss $\mathcal{L}_{Per}$, GAN loss $\mathcal{L}_{GAN}$, reconstruction loss $\mathcal{L}_{Recon}$ for render network training. To enhance the quality of mouth region, we obtain the mouth mask and

introduce a mouth region perceptual loss $\mathcal{L}_{Per}^{Mouth}$. The overall training objective is formulated as follows:

$$\mathcal{L} = \mathcal{L}_{Per} + \mathcal{L}_{GAN} + \mathcal{L}_{Recon} + \mathcal{L}_{Per}^{Mouth} \tag{10}$$

During inference, the renderer generates the final video by projecting the visual codes predicted by OmniTalker onto 3D keypoints through 3DMM.

## C  Experiment Details

### C.1  Datasets

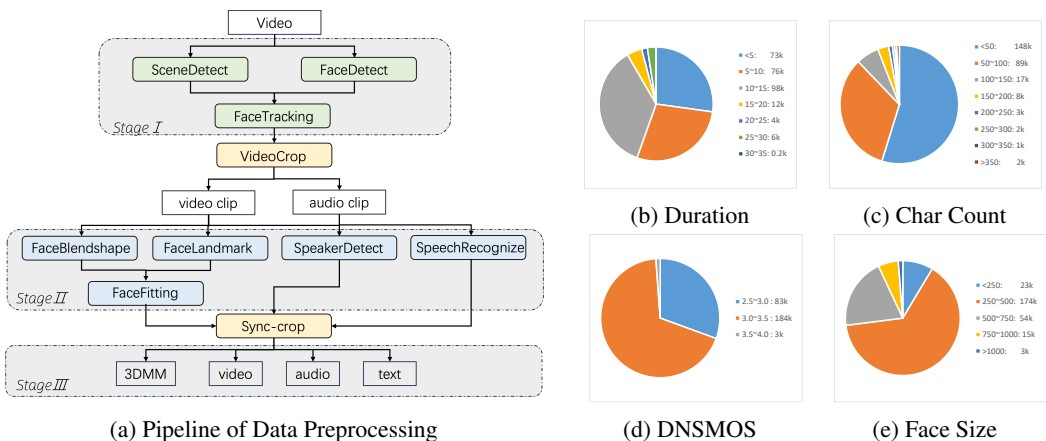

(a) Pipeline of Data Preprocessing     (b) Duration     (c) Char Count     (d) DNSMOS     (e) Face Size

Figure 5: Overview of our proposed pipeline and dataset.

We employ TalkingHead-1KH [52], VoxCeleb [36], and CelebV-HQ [65] as pre-training datasets, while constructing a high-quality custom dataset for fine-tuning. This study proposes an automated preprocessing pipeline for large-scale multimodal data curation, as illustrated in Figure 5a. The pipeline comprises three sequential stages, containing eight specialized processing modules strategically designed to enable parallel execution and optimize computational efficiency through modular architecture.

- Stage 1: Coarse Segmentation
    - Scene Detection: PySceneDetect [10] with adaptive threshold ($\sigma = 1.5$)
    - Face Detection & tracking: Insightface [17] with IoU continuity $> 0.5$
- Stage 2: Multimodal Feature Extraction
    - Facial motion: FaceVerse [49] (52 blendshapes + 6DOF pose + 4 eye gaze)
    - Speaker Detection: LightASD [29] (confidence $> 0.8$)
    - Speech Recognition: Whisper-V3-Large [38]
- Stage 3: Fine Segmentation
    - Temporal constraints: Clip duration: $1s \leq t \leq 30s$ & Phoneme rate: $< 1s/character$
    - Spatial constraints: Face bounding box $> 15\%$ frame area
    - Others: Audio quality: DNSMOS P.835 OVRL [39] (cutoff $> 2.5$)

Figures 5b to 5e shows the distributions across duration, text length, quality score, and average face size in custom dataset. It comprises approximately 300,000 clips, totaling 690 hours of high-quality multimodal data with synchronized text, audio, and video components.

### C.2  Training

**Strategy.**  We begin by performing large-scale pre-training on open-source multimodal datasets as described in Appendix C.1 to establish foundational capabilities in text comprehension and

multimodal generation. During the training process, we implement a masking strategy that randomly masks 70-100% of the audio-visual sequences, compelling the model to learn sequence reconstruction abilities. Concurrently applying random dropout with a 0.2 probability across text, audio, and video conditioning inputs facilitates classifier-free guidance (CFG) training. In the final stage, we conduct fine-tuning on custom dataset, which significantly enhances the model's performance.

**Sampling.** We modified the timestep sampling distribution from a uniform distribution to a logit-norm distribution (mean=0.0, std=1.0) according to [13], and observed that employing logit-norm bias to prioritize the selection of training timesteps significantly enhances model performance than uniform distributions.

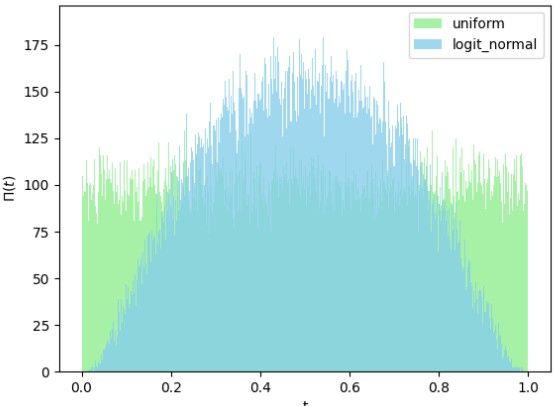

Figure 6: logit-norm and uniform distributions that bias the sampling of training timesteps.

## C.3 More Qualitative Evaluation

**Eye Movements.** Figures 9 and 10 demonstrate the enhanced performance of our proposed method in generating eye movements. As indicated by the highlighted red regions in the visualizations, our approach produces complex gaze trajectories with dynamic variations that exhibit significant temporal correlations with head pose changes. By contrast, the baseline method maintains a fixed gaze direction determined at initialization, resulting in nearly static eye movements. This coordinated gaze-head

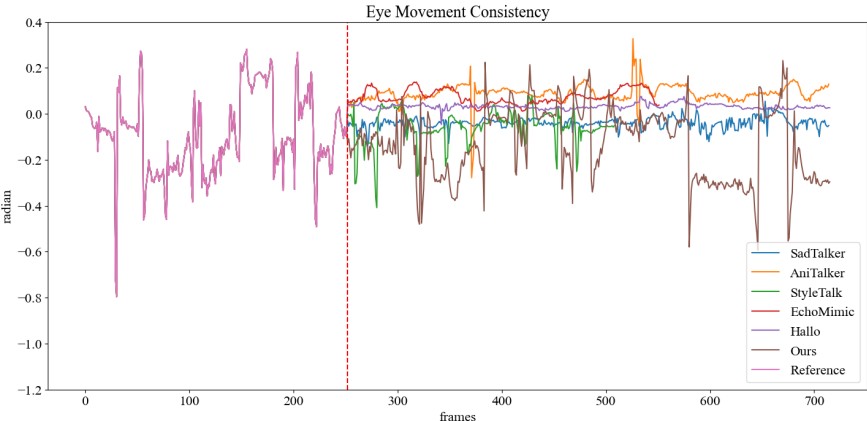

Figure 7: Visualization of eye movements over time for the reference video and videos generated by different methods. The red dashed line separates the values of the reference video on the left from those of the generated results on the right.

motion mechanism plays a critical role in enhancing the biological plausibility of generated videos. Notably, Figure 7 further validates our method's capability to inherit eye movement characteristics from reference videos. Experimental results show that the generated gaze trajectories demonstrate heritable features in both frequency and amplitude parameters compared to reference videos, while maintaining necessary difference to avoid mechanical repetition while preserving consistency.

**Emotional Generation.** As illustrated in Figure 8, we employ reference videos from the RAVDESS[32] emotional dataset. Leveraging in-context learning capabilities, our methods enables to generate results highly aligned with target emotions. Experimental results demonstrate that the proposed method accurately captures expressive facial micro-expressions. This capability of generating multimodal emotional representations effectively validates the model's capacity for analyzing and reconstructing complex emotional states.

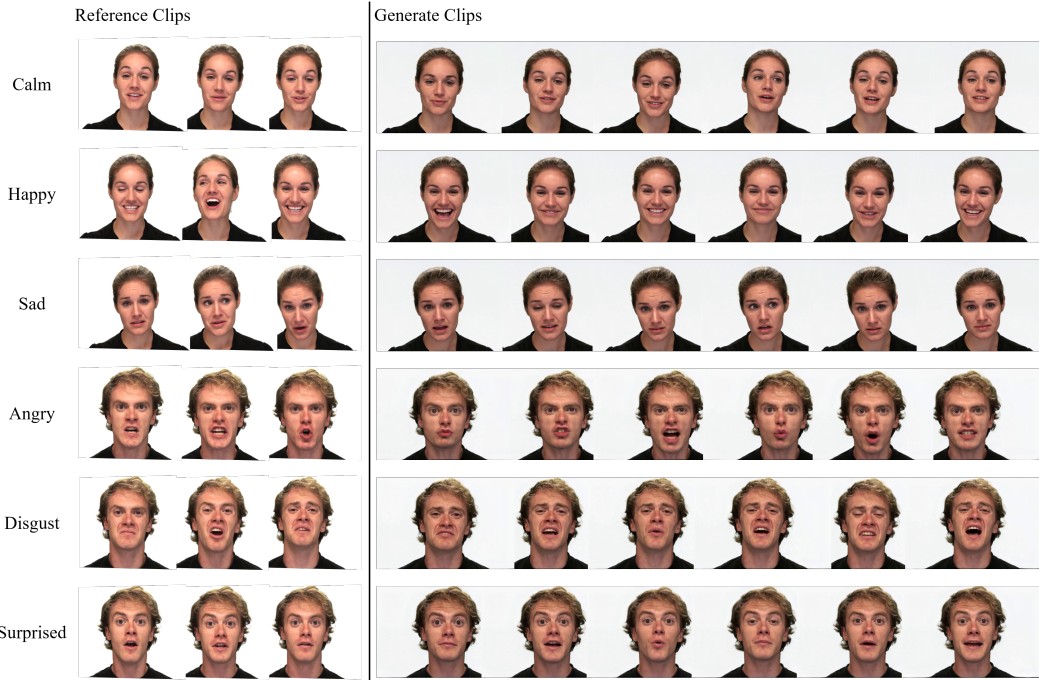

Figure 8: Visualization of Generation via different emotional reference.

## C.4 User Study Settings

We conducted user studies for both audio and video generation following our quantitative evaluation. We selected 25 reference videos each from the VoxCeleb2 and Custom datasets, totaling 50 references. For the audio evaluation, we generated 50 audio samples per TTS method using the provided text, then employed our synthesized audios to generate corresponding videos for each THG method. Each participant was asked to evaluate the results across six dimensions: for audio outputs, speech similarity(assessing vocal timbre resemblance) and speech rhythm(evaluating prosody alignment with text punctuation); for video outputs, visual quality(focusing on image clarity, artifacts, and geometric distortion), speaking style(comparing emotional expression, head motion, and gaze patterns), lip-sync, and pose-sync(verifying semantic accuracy of head gestures like nodding for affirmation). Figure 11 displays the interface used in our user study. 50 participants rated each generation result on a 5-point scale (1-5), with average scores presented in Table 3. This comprehensive evaluation framework ensures multi-dimensional assessment of both audio-visual quality and semantic fidelity in human-like generation tasks.

## C.5 More Ablation Study

We have conducted additional experiments to validate the efficacy of the Dual-Branch DiT architecture, and the preliminary results align with our original conclusions.

Table 7: Modality ablation experiments

| Setting | WER↓ | SIM-A↑ | Sync-C↑ | E-FID↓ | P-FID↓ |
|---|---|---|---|---|---|
| text only $(\alpha_{M_r} = 0.0, \alpha_{C_r} = 0.0, \alpha_{S_T} = 2.0)$ | 1.65 | 0.249 | 4.66 | 0.87 | 4.21 |
| w/o audio $(\alpha_{M_r} = 0.0, \alpha_{C_r} = 2.5, \alpha_{S_T} = 2.0)$ | 1.65 | 0.264 | 4.45 | 0.12 | 1.67 |
| w/o video $(\alpha_{M_r} = 2.0, \alpha_{C_r} = 0.0, \alpha_{S_T} = 2.0)$ | 1.61 | 0.771 | 4.93 | 0.32 | 1.88 |
| full $(\alpha_{M_r} = 2.0, \alpha_{C_r} = 2.5, \alpha_{S_T} = 2.0)$ | 1.58 | 0.762 | 5.48 | 0.08 | 0.12 |

Table 7 compares the performance of four different settings. The full model (WER=1.58, Sync-C=5.48) outperforms all ablated variants, demonstrating that cross-modal integration of audio and video enhances text-to-speech generation accuracy and improves temporal alignment between modalities. When reference audio is not provided, the model cannot replicate the original speaker's vocal characteristics, as evidenced by a significant drop in SIM-A. However, the generated lip movements and head motions align naturally with the input text. Without reference video input, lip movements and head motions are primarily driven by audio loudness and textual content. Although synchronization remains coherent, person-specific details (e.g., unique facial expressions) are not preserved, resulting in significantly lower E-FID and P-FID scores. When only text is provided (without reference audio or video), the model stochastically generates a plausible speaking style, encompassing vocal tone, rhythm, emotional expression, lip motion, and head motion patterns. While speech articulation remains clear, the specific identity and style vary randomly. This highlights the effectiveness of the dual-branch architecture in enabling robust cross-modal learning and replication capabilities.

# D Limitations and Future Work

Here we discuss the limitations of this work and potential directions for future improvement. Firstly, the scope of our method focuses primarily on dynamic generation of the head region without incorporating hand pose modeling or full-body motion control. This constraint limits the completeness and interactivity of generated content to some extent. The second challenge relates to generation quality: GAN-based approaches still face technical barriers when handling large-scale dynamic transformations. Notably, when motion amplitude exceeds critical thresholds, artifacts such as texture blurring and boundary discontinuities frequently emerge, compromising the visual realism of generated results.

To address these limitations, we propose optimizations along two dimensions. First, in the motion control dimension, we can implement a coordinated generation framework by integrating hand skeletal tracking data with full-body pose estimation information. Second, for generation quality enhancement, we recommend adopting a video diffusion model-based generation paradigm. Compared to traditional GAN architectures, video diffusion models demonstrate superior spatiotemporal continuity through their progressive denoising process. The progressive sampling mechanism effectively mitigates quality degradation caused by large-scale movements. Furthermore, combining attention mechanisms with spatiotemporal feature fusion strategies promises to significantly enhance both detail fidelity and dynamic smoothness in generated outputs.

# E Broader Impacts

Our work focuses on the efficient generation of realistic and expressive digital human videos, aiming to drive technological advancement with positive societal impact. By leveraging automation, real-time interaction, and multimodal perception capabilities, this technology enhances productivity across industries and accelerates innovation in the integration of text, audio, and video processing systems.

It also fosters the emergence of new industries such as virtual idols and digital avatars, while enabling cross-lingual content generation that promotes cultural exchange across regions. However, the same technology carries risks: hyper-realistic deepfake algorithms could be exploited for disinformation campaigns, identity fraud, or synthetic media manipulation; sensitive biometric data collection required for digital humans may lead to large-scale privacy breaches if not properly secured; and prolonged engagement with virtual companions or personalized avatars might inadvertently create emotional dependencies or psychological impacts. To address these challenges,

- We incorporate visible watermarks into generated videos to proactively alert users that the content is synthetic in nature.
- We embed imperceptible digital watermarks within both video and audio streams to enable traceability and track the origin of generated content, thereby requiring creators to consider potential legal and ethical risks associated with synthetic media production.
- We are committed to advancing our methodology to enhance deepfake detection techniques, aiming to improve the accuracy and reliability of automated detection systems through continuous algorithmic refinement.

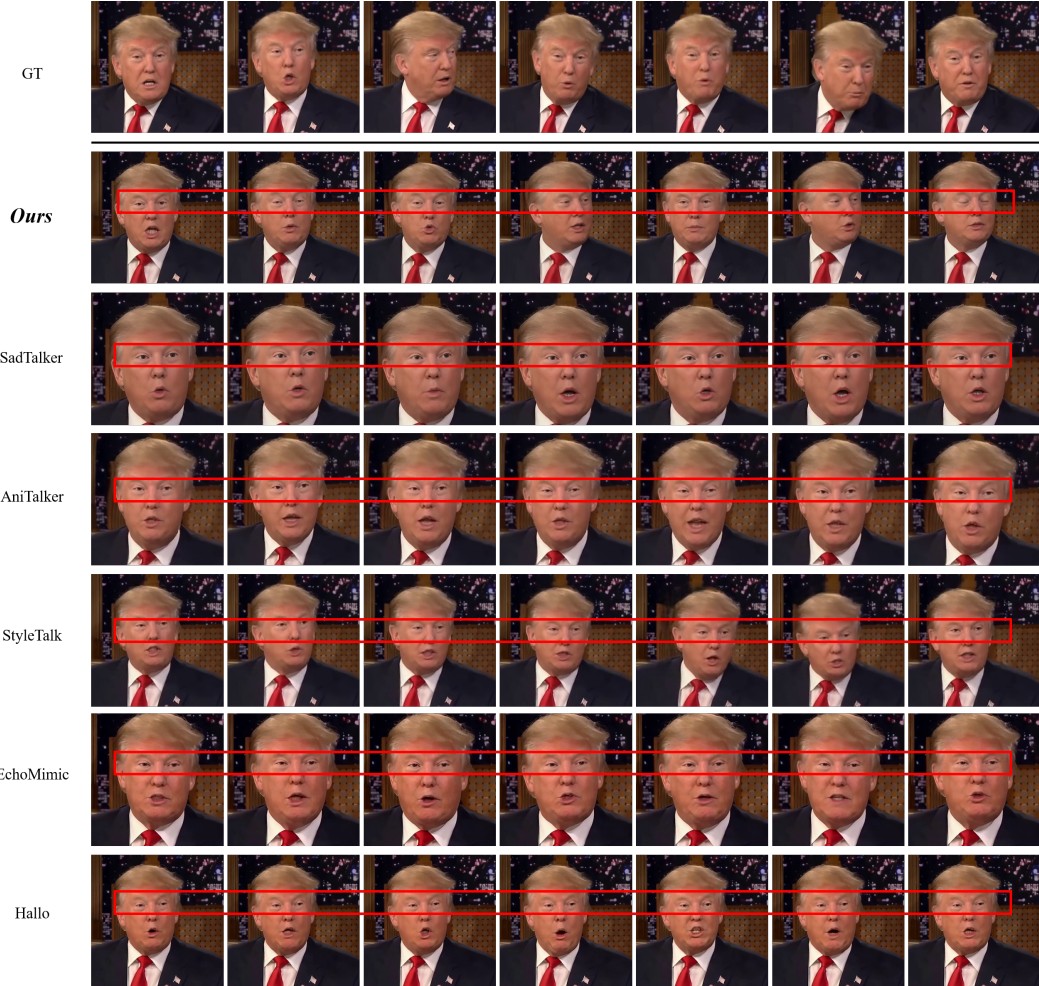

Figure 9: Comparative Analysis of Eye Movement Visualization on ID1.

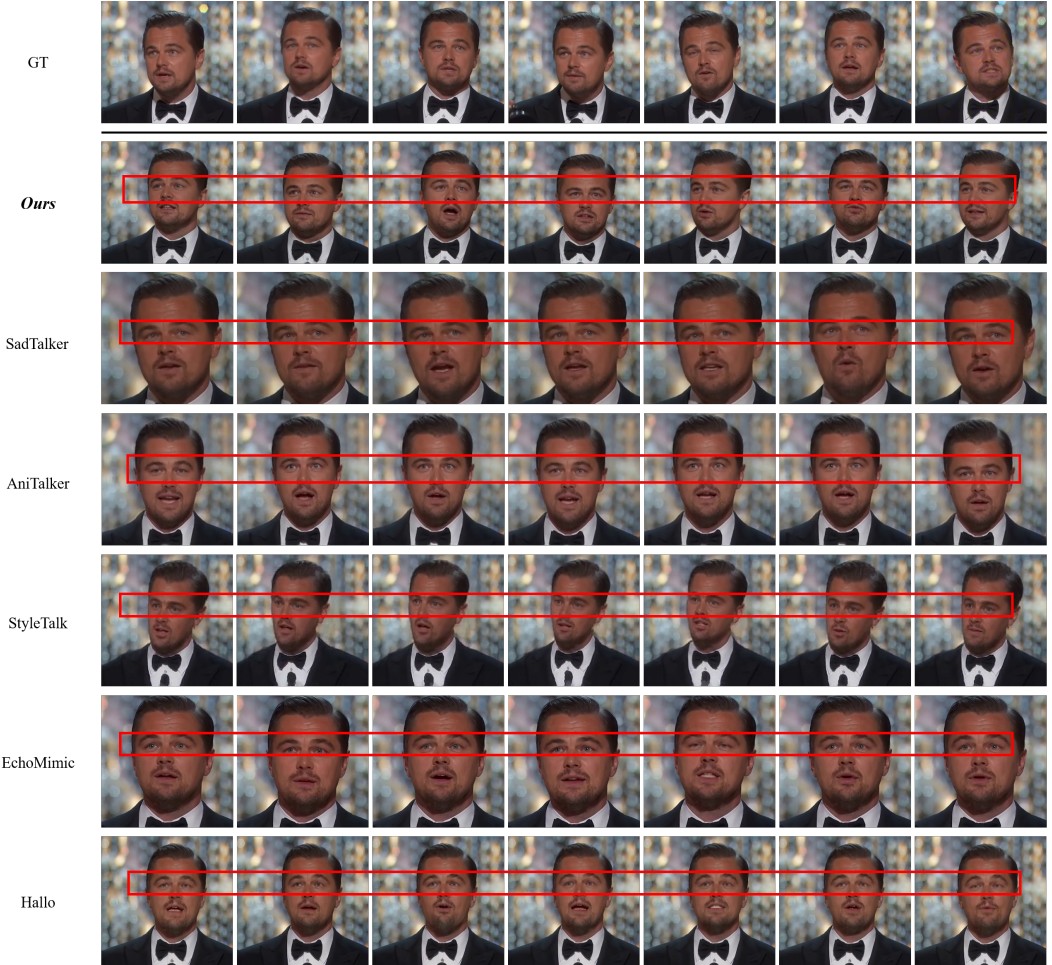

Figure 10: Comparative Analysis of Eye Movement Visualization on ID2.

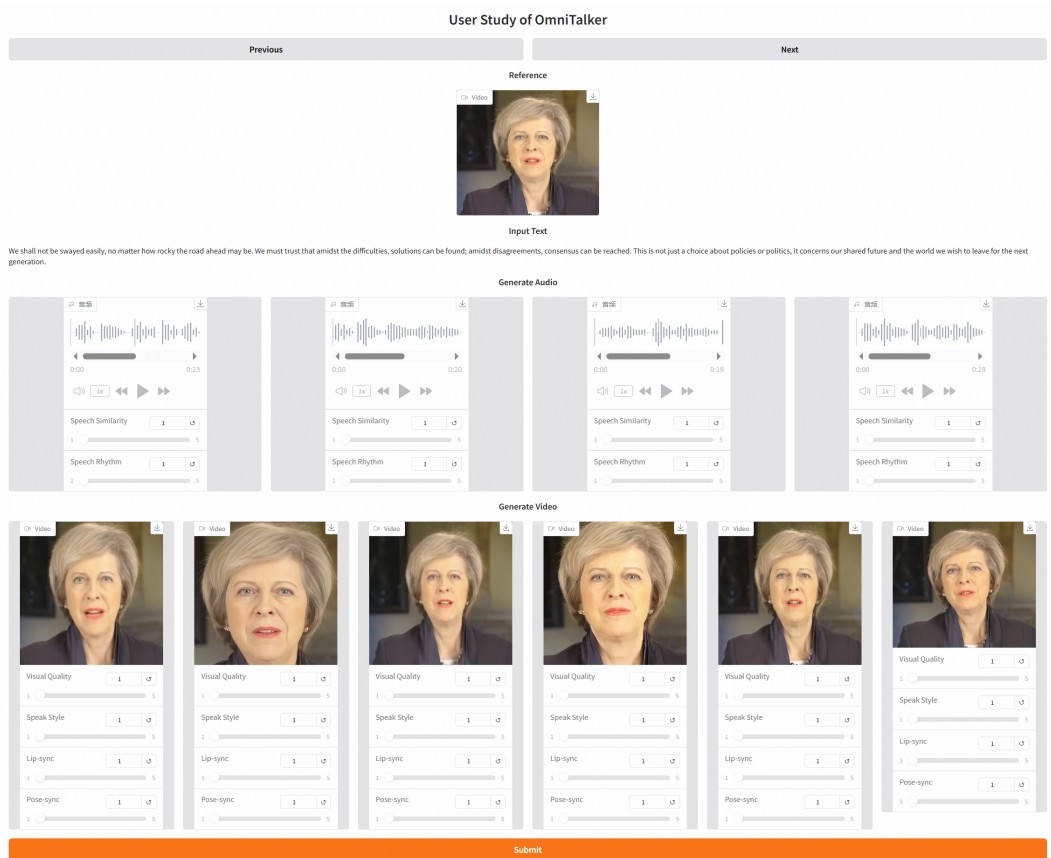

Figure 11: The interface of user study.

