# OpenReview forum: "OmniTalker: One-shot Real-time Text-Driven Talking Audio-Video Generation With Multimodal Style Mimicking"
_NeurIPS.cc/2025/Conference — NeurIPS 2025 poster_

### Official Review · Reviewer_7PYd · 2025-06-25

**Clarity:** 2
**Significance:** 3
**Originality:** 3
**Rating:** 5
**Confidence:** 3

**Summary:**

This paper introduces OmniTalker, the first unified framework for one-shot real-time talking head generation based on text. By developing a DiT-based backbone with audio-video fusion, OmniTalker generates high-quality synchronous video and audio that mimics the styles of the reference. The contributions include a dual-branch DiT-based backbone, a unified framework for training and one-shot generation, and a whole system that could generate at a high frequency of 25Hz. The framework is evaluated on publicly available datasets and custom datasets, compared with TTS and audio-driven methods, and shows improvements. This work provides a prototype for real-time text-driven talking head generation and provides further study directions.

**Questions:**

1.  Could the authors clarify whether they have a plan to release the custom dataset?  This will increase the soundness and the influence of this work.

2. In Table 1, OmniTalker underperforms previous work on SEED dataset. Could the authors give a further discussion on this? Additionally on Custom dataset, OmniTalker gives the best result. It is explained in the paper that the video could also help improve audio performance. However, why  SIM-A with $\alpha_{C_r}=2.5$ is worse on the Custom dataset?

3. In Related Work section, line 62, it is said a detailed comparison with prior methods is included in the Appendix. However the paper do not include an appendix and I did not find it in the supplementary material as well.

4. How is the hyperparameters, like block size, determined?

**Ethical Concerns:**

["NO or VERY MINOR ethics concerns only"]

**Final Justification:**

My final recommendation for this paper is acceptance. All my questions are addressed by the authors.
### Q1. Question about whether the authors will release the custom dataset
The authors promise to release their dataset in 2025 Q4, and provided a timeline for this.
### Q2. Question about the performance of OmniTalker
The authors explained the reasons and future research plans.
### Q3. Question about missing section.
This section is added in the revised manuscript and in comment.
### Q4. Question about hyperparameters.
This is explained in the response.

**Limitations:**

yes

**Quality:**

2

**Strengths And Weaknesses:**

Strengths:
1. This work is the first one-shot real-time text-driven talking head generation framework and fill the gap in this area.
2. The entire pipeline of the framework is well demonstrated, providing a clear methodology for further research in this area.

Weakness:
1. Lack of comparison with existing text-driven THG baselines (e.g., TTSF [1] referred to in related works).
2. The custom dataset is not available, and the results are hard to check. Although the authors claim the availability of the dataset, I did not find a link to the dataset.
3. The ablation is not adequate, and the effectiveness of CFG and dual-branch DiT is not fully validated.

[1] Jang, Youngjoon, et al. "Faces that speak: Jointly synthesising talking face and speech from text." Proceedings of the IEEE/CVF Conference on Computer Vision and Pattern Recognition. 2024.

---

> ### Author Rebuttal · Authors · 2025-07-31
>
> # Response to Reviewer 7PYd
>
> Thank you for your valuable feedback and suggestions. We have prepared individual responses to each of your questions.
>
> ## Q1
>
> Yes, we have plans to release the custom dataset. We are currently finalizing the necessary steps, including ethical reviews and data curation, and aim to make it publicly available by Q4 2025.
>
> ## Q2
>
> Thank you for your insightful questions. We appreciate the opportunity to elaborate on these points:
> 1. **WER performance.** Our training dataset maintains a 3:1 ratio of Mandarin to English data. This imbalance likely contributes to the disparity in performance between SEED-ZH/Custom (where we achieve the best WER) and SEED-EN (where F5-TTS outperforms us). The limited English data constrains the model's ability to generalize to English pronunciation patterns, particularly in challenging acoustic scenarios. We are actively exploring data augmentation and rebalancing strategies to address this limitation in future work.
> 2. **SIM-A performance.** The suboptimal SIM-A performance arises from two factors: (1) the training dataset contains regional accents and non-standard pronunciations, which challenge both acoustic modeling and timbre preservation. Our model's reliance on pinyin or character-level alignments may create bottlenecks in capturing these variations, especially when they deviate from canonical pronunciation rules. (2) We found that during early-stage training, the model tends to prioritize speaker identity matching (SIM-A), as coarse timbre matching is simpler than learning precise text-to-speech alignments. However, in the mid-training phase, as the model refines text-to-speech alignment, its focus shifts away from optimizing SIM-A, causing this metric to plateau. Additionally, while video integration improves prosody, it introduces gradient conflicts. The model prioritizes visual-audio synchronization over fine-grained timbre preservation, resulting in a slight degradation in SIM-A performance.
>
> We have retrained our model for an additional 500k steps and observed a 0.02 improvement in SIM-A on the Custom dataset. User studies confirm enhanced accent consistency, suggesting that the original performance was primarily due to insufficient training convergence rather than a fundamental limitation of the model architecture.
>
> We have conducted additional experiments to validate the efficacy of the Dual-Branch DiT architecture, and the preliminary results (shown in the attached table) align with our original conclusions. These will be included in the revised paper.
>
> | Setting | WER $ \downarrow $ | SIM-A $ \uparrow $ | Sync-C $ \uparrow $ | E-FID $ \downarrow $ | P-FID $ \downarrow $ |
> | --- | :---: | :---: | :---: | :---: | :---: |
> | Text only ($ \alpha_{M_r}=0.0 $,$ \alpha_{C_r}=0.0 $,$ \alpha_{S_T}=2.0 $) | 1.65 | 0.249 | 4.66 | 0.87 | 4.21 |
> | w/o audio  ($ \alpha_{M_r}=0.0 $,$ \alpha_{C_r}=2.5 $,$ \alpha_{S_T}=2.0 $) | 1.65 | 0.264 | 4.45 | 0.12 | 1.67 |
> | w/o video  ($ \alpha_{M_r}=2.0 $,$ \alpha_{C_r}=0.0 $,$ \alpha_{S_T}=2.0 $) | 1.61 | **0.771** | 4.93 | 0.32 | 1.88 |
> | Full  ($ \alpha_{M_r}=2.0 $,$ \alpha_{C_r}=2.5 $,$ \alpha_{S_T}=2.0 $) | **1.58** | 0.762 | **5.48** | **0.08** | **0.12** |
>
>
> The table above compares the performance of four different settings. The full model (WER=1.58, Sync-C=5.48) outperforms all ablated variants, demonstrating that cross-modal integration of audio and video enhances text-to-speech generation accuracy and improves temporal alignment between modalities. When reference audio is not provided, the model cannot replicate the original speaker's vocal characteristics, as evidenced by a significant drop in SIM-A. However, the generated lip movements and head motions align naturally with the input text. Without reference video input, lip movements and head motions are primarily driven by audio loudness and textual content. Although synchronization remains coherent, person-specific details (e.g., unique facial expressions) are not preserved, resulting in significantly lower E-FID and P-FID scores. When only text is provided (without reference audio or video), the model stochastically generates a plausible speaking style, encompassing vocal tone, rhythm, emotional expression, lip motion, and head motion patterns. While speech articulation remains clear, the specific identity and style vary randomly. This highlights the effectiveness of the dual-branch architecture in enabling robust cross-modal learning and replication capabilities.
>
>
> ## Q3
>
> Thank you for pointing out the omission in Sec. 2. We sincerely appreciate your careful reading and constructive feedback. We will add a detailed discussion in Appendix in the revised manuscript as follows:
>
> | Method | text driven | one-shot | voice clone | style replicate | real-time |
> | --- | :---: | :---: | :---: | :---: | :---: |
> | Wav2Lip | | &#10003; | | | &#10003; |
> | VividTalk | | &#10003; | | | &#10003; |
> | SadTalker | | &#10003; | | | &#10003; |
> | StyleTalk | | &#10003; | | &#10003; | |
> | VASA-1 | | &#10003; | | | &#10003; |
> | EchoMimic | | &#10003; | | | |
> | Hallo | | &#10003; | | | |
> | TTSF | &#10003; | &#10003; | | | &#10003; |
> | AV-Flow | &#10003; | | &#10003; | &#10003; | &#10003; |
> | **Ours** | &#10003; | &#10003; | &#10003; | &#10003; | &#10003; |
>
>
> ## Q4
>
> In determining the model scale, we primarily consider two critical factors: model capacity and the computational requirements dictated by real-time performance constraints. Considering the data scale, the input text has a vocabulary size of approximately 2.5k, while the audio/visual representations consist of 100D mel-spectrograms and 60D motion codes. To balance model capacity with real-time inference efficiency, we set the hidden dimensions to 512/1024 for text/audio-visual embeddings. Building upon insights from state-of-the-art TTS models like F5-TTS, the architecture is carefully calibrated to optimize parameter efficiency within a total parameter budget of approximately 0.8 billion, achieving a balanced trade-off between computational resources and performance.
>
> We regret any lack of clarity in our initial submission due to the constraints of space and the volume of revisions. We will incorporate all remaining suggestions in our final submission.

---

> > ### Comment · Reviewer_7PYd · 2025-08-05
> >
> > I thank the authors for the response. My concerns are addressed.

---

> > > ### Author Response · Authors · 2025-08-08
> > >
> > > We are deeply grateful to you for your time and effort in evaluating our work. Your feedback has strengthened the manuscript significantly, and we hope the revisions meet your expectations. We look forward to any further suggestions and are confident that the revised version will contribute meaningfully to the field.

---

### Official Review · Reviewer_Xaqr · 2025-06-26

**Clarity:** 2
**Significance:** 3
**Originality:** 3
**Rating:** 5
**Confidence:** 4

**Summary:**

This paper introduces a new setting: given a reference video and a driving text prompt, the model can generate synchronized audio-visual talking heads while preserving the speaking style (both vocal and facial) of the reference video. To achieve this, the authors first employ a shared encoder to encode the text prompt, reference video, and audio, followed by two separate heads dedicated to generating audio and video, respectively. Additionally, they adopt a masked infilling strategy during training, which helps maintain each individual’s unique speaking style during generation. Notably, the proposed approach is capable of running in real time on a single GPU, and demonstrates strong audio-visual synchronization performance.

**Questions:**

1. How does this method perform in singing scenarios?
2. Is it possible to generate results without a reference video? What happens if only audio or video is provided? Can the system still generate plausible outputs in those cases?
3. How is the renderer trained? Are the 3D keypoints you use the commonly adopted 68 facial landmarks? If so, how do you ensure the motion consistency of features like hair and shoulders, which are not explicitly covered by those keypoints?
4. Does the reported 24 FPS include the 3D face reconstruction step?
5. Since your rendering module essentially operates as an image-based model, and the reference image is randomly selected, how do you ensure the consistency of features like teeth, which may not be clearly visible in all reference images?
6. Given that many video generation methods based on Stable Diffusion tend to degrade when generating long videos due to overlapping inference artifacts, can this method generate long videos without such degradation?

**Ethical Concerns:**

["NO or VERY MINOR ethics concerns only"]

**Final Justification:**

The author's response addresses some of my concerns. I decide to maintain my original score.

**Limitations:**

yes

**Quality:**

3

**Strengths And Weaknesses:**

Strength:
1. They are the first to introduce a novel setting: given a reference video and a driving text, the system is able to generate synchronized audio-visual talking heads.
2. The generated video preserves the speaking style of the reference, including both vocal characteristics and facial expressions.
3. The proposed method achieves real-time inference at 25 FPS on a single GPU.
4. The generated results exhibit high-quality audio-visual synchronization.


Weakness:
1.  This paper uses a 3DMM model as the motion representation, which inherently limits the method to the scope of 3DMM. For example, it can only handle real human faces, and lacks the flexibility of former diffusion-based approaches （\eg, ominihuman, loopy) that can generalize to a wider range of scenarios, such as AIGC-generated avatars, cartoons, or animals. Additionally, 3DMM reconstructions can become inaccurate under large head poses.
2. The paper’s claim of being end-to-end seems somewhat misleading. In fact, the audio and video generators are first pre-trained separately, and the DiTs are only used later to generate intermediate representations. This setup is quite similar to previous methods such as Float and VASA-1.
3. The paper lacks comparisons with other real-time and open-source methods, such as JoyVASA and DiTTO, which should be included for a more comprehensive evaluation.
4. Some training details are missing. For instance, how is the renderer trained? What exactly do the “3D keypoints” represent? Since 3DMM only accounts for the facial region, how is the dynamic behavior of hair ensured during generation?

---

> ### Author Rebuttal · Authors · 2025-07-31
>
> # Response to Reviewer Xaqr
>
> Thank you for your expert feedback and suggestions. We have carefully considered each of your points and provided responses accordingly.
>
> ## Q1
>
> This is a highly relevant question! We have indeed experimented with using singing segments as reference clips, but the results are not yet satisfactory. While the model maintains strong capabilities in timbre cloning and phonetic clarity (accurately preserving vocal characteristics and text alignment), it faces significant challenges in generating musically coherent outputs. Generated results often exhibit fragmented rhythms, unnatural chord progressions, and mismatched background music (BGM), leading to a lack of harmonic cohesion between vocals and accompaniment.
> On the one hand, our training datasets lack high-quality singing samples with accompaniment. Audio containing background music often scores poorly on DNSMOS (a speech quality metric) and was filtered out during preprocessing according to Appendix B.1. This limits the model's  ability to learn musical structures like melody-lyric alignment and rhythmic phrasing. On the other hand, singing scenarios demand modeling multi-dimensional dependencies (text, pitch, tempo, BGM), which is far more complex than text-to-speech (TTS). With only reference audio and text as inputs, the model struggles to infer musical intent (e.g., melody direction, harmonic context) without explicit rhythmic or BGM guidance.
> To address these challenges, we plan to:
> 1. **Add Musical Control Signals**. Introduce explicit inputs like tempo (BPM), rhythmic patterns, chord sequences, or MIDI-like annotations to guide musical structure generation.
> 2. **Expand Training Data**. Collect a singing-TTS dataset with multimodal annotations (e.g., melody contours, rhythm labels) to strengthen supervision on musical elements.
> 3. **Explore staged training**. First optimizing vocal content (timbre, lyrics, pitch) and then refining background music generation separately to improve overall coherence.
>
> While singing generation is not currently prioritized in our open release, we are actively iterating on this capability. We look forward to sharing progress soon!
>
> ## Q2
>
> That's an excellent question that highlights additional user cases. Thanks to our training strategy as mentioned in Appendix B.2, which involves randomly dropping text, audio, or video conditioning inputs with a certain probability, our model inherently supports unconditional generation. Here's how it works in different scenarios:
> 1. **Text-only input**. When only text is provided (without reference audio/video), the model stochastically generates a plausible speaking style, including vocal tone, rhythm, emotional expression, lip motion, and head motion patterns. While the speech articulation remains clear, the specific identity and style will vary randomly.
> 2. **Audio-only input**. With only audio input, the lip movements and head motions are primarily influenced by the audio loudness and text content. While the synchronization stays coherent, person-specific details (e.g., unique facial expressions) will not be preserved.
> 3. **Video-only input**. When only video is provided, the model cannot replicate the original speaker's vocal characteristics. However, the generated lip movements and head motions will align naturally with the input text.
>
> In all cases, the outputs maintain logical coherence and visual/audio alignment, though specific identity traits may be lost when certain modalities are absent. This flexibility enables diverse applications while preserving core functionality.
>
> ## Q3
>
> The renderer we use is a warp-based GAN generator trained via a self-reenactment framework as described in Appendix A.2. Specifically, we first extract the appearance features from source image, and then leverage 3D facial landmarks to drive the transformation of expression and pose from the source to the target through a warping module. The final image is synthesized by a SPADE-based generator that effectively fuses the warped features with spatially adaptive normalization.
>
> Regarding the 3D landmarks, we manually select a set of key points from the 3DMM model that are representative of facial expressions and head pose—these are not limited to the standard 68 2D facial landmarks, but are instead chosen to optimally capture 3D facial dynamics. While these keypoints primarily correspond to facial structures and do not explicitly cover regions such as hair or shoulders, the network learns to implicitly extend the influence of each keypoint during training. This allows nearby non-facial regions (e.g., hair and upper shoulders) to be co-driven in a spatially coherent manner, ensuring motion consistency across the entire head and surrounding areas. The learned deformation field effectively generalizes the control beyond the immediate vicinity of the keypoints, enabling natural and holistic head motion rendering.
>
> ## Q4
>
> The reported 24 FPS does not include the 3D face reconstruction step. Given a reference video, we first perform 3D face reconstruction as a preprocessing step, which is executed only once. During the _inference_ phase, instead of conducting reconstruction, we utilize the 3DMM model to convert the output motion blendshape parameters into facial landmark coordinates. These landmarks serve as control signals for the subsequent GAN-based model. The vertex coordinate calculations involved in this conversion process incur negligible computational overhead on the GPU.
>
>
> ## Q5
>
> We would like to clarify the use of randomly selected reference images in our rendering framework. The random selection strategy is primarily employed during the _training_ phase of the renderer to enhance the model's robustness and generalization capability. Indeed, we observe that the choice of reference image—particularly whether it shows teeth or open eyes—can influence the visual details in the generated output.
>
> However, during the _inference_ stage in practical application, we do not select reference images randomly. Instead, we adaptively choose a reference frame from the input reference video based on facial expression analysis. Specifically, to ensure consistent and realistic rendering of features such as teeth, we select a reference image in which the subject's mouth is open and teeth are clearly visible. This strategy ensures that the renderer produces temporally coherent and identity-consistent results, including accurate appearance of internal facial structures like teeth, while maintaining natural motion dynamics.
>
>
> ## Q6
>
> Great question！Diffusion-based models employ a fixed-length generation mechanism. To ensure pixel-level continuity across different inference clips, they require not only appearance reference frames but also temporal reference frames. However, this temporal reference frame integration can lead to error accumulation during long-sequence inference (because of the gap of temporal reference frames between _training_ and _inference_ phases). Instead of directly generating pixels, our diffusion model only generates mel-spectrograms (for audio) and motion codes (for video), which allows for seamless transitions via simple fade-in/fade-out smoothing rather than complex pixel-level alignment. The GAN-based renderer handles pixel generation by warping the appearance of reference frames according to motion codes frame-by-frame without temporal reference frames during _inference_. This design decouples audio-visual coherence from pixel complexity, enabling scalable long video synthesis while maintaining fidelity. We successfully generated 30-minute videos with consistent quality, showing no visible artifacts or quality loss.
>
> While we have addressed each of your questions in our responses, we must apologize for any remaining ambiguities due to the limited space. Please rest assured that we will incorporate all remaining suggestions in our final submission.

---

> > ### Comment · Reviewer_Xaqr · 2025-08-07
> > **Official Comment by Reviewer Xaqr**
> >
> > Thanks for your detailed clarifications, which address some of my concerns. I decide to maintain my original score.

---

> > > ### Author Response · Authors · 2025-08-08
> > >
> > > We are very appreciated for your expert review and detailed suggestions. Your recommendations have led to substantial improvements in our manuscript, and we hope the revised version meets your expectations. We would be grateful for any further observations and expect this refined work will advance discussions within the academic community.

---

### Official Review · Reviewer_NKZk · 2025-06-30

**Clarity:** 3
**Significance:** 3
**Originality:** 3
**Rating:** 5
**Confidence:** 5

**Summary:**

This paper presents OmniTalker, a novel and unified framework for generating synchronized audio and video of a talking head from text input. The method is one-shot, meaning it can mimic a person's identity and speaking style from a short reference video. The core of the architecture is a dual-branch Diffusion Transformer (DiT) that jointly generates audio (mel-spectrogram) and visual motion features. The key innovation is the use of a masked infilling strategy, which allows the model to learn multimodal styles (vocal characteristics, head motion, facial expressions) in-context, without requiring an explicit style encoder.
The paper is exceptionally strong, presenting a well-motivated and elegant solution to a challenging problem. The experimental results are comprehensive and demonstrate state-of-the-art performance, particularly in style preservation, where it vastly outperforms existing methods. The work is well-written, the architecture is clearly explained, and the claims are robustly supported by quantitative metrics, qualitative examples, and a user study. This is a high-quality submission that represents a significant advance in the field of talking head generation.

**Questions:**

1. Could you clarify the exact input representation for the text projector? Does the model use raw text, or an intermediate format like phonemes? Given the challenge of generating mel-spectrograms directly from text, more detail on this part of the pipeline would be very helpful.
2. Did you consider an end-to-end approach that generates video pixels directly, instead of intermediate motion representations? My concern is that the GAN-based renderer might be tightly coupled with the learned motion codes, potentially limiting their applicability with different rendering architectures and may perform worse when applying a difference alignment method to input video.
3. Did you apply another pre-trained weights when you perform large-scale pre-training, or just train from scatch?

**Ethical Concerns:**

["NO or VERY MINOR ethics concerns only"]

**Final Justification:**

Based on its novelty and the interesting nature of the task, I recommend acceptance.

**Limitations:**

yes

**Paper Formatting Concerns:**

None.

**Quality:**

3

**Strengths And Weaknesses:**

Strengths:
1. The authors proposed an elegant method to solve text-driven talking head generation task in a unified, end-to-end framework that jointly synthesis both modalities.
2. The proposed method is effective, which show really good visually results that achieve high lip-sync and high rendering quality simultaneously, and achieve comparable quality in text-to-speech results with the sota tts methods.
3. The proposed method is one-shot, only a reference video and a reference audio needed to generate same style results.
4. The paper is well-writen and easy to follow.

Weaknesses:
1. The generated video and audio are rely on off-the-shelf video render and audio vocoder, which limite the output quality to upper-bound of the decoder they adopt.

---

> ### Author Rebuttal · Authors · 2025-07-31
>
> # Response to Reviewer NKZk
>
> Thank you for your careful reading, helpful comments, and constructive suggestions. We have responded to each question individually.
>
> ## Q1
> That is a good question！As we stated in Sec. 3.1(Line 127)，the driving text and reference text undergo a unified tokenization process: Chinese text is converted into pinyin sequences, while Latin-based languages are represented as character/letter sequences. Specifically, we maintain a vocabulary list that includes symbols, digits, characters, pinyin sequences for Chinese characters, and additional elements (an illustrative example of a vocabulary list: [!, ", #, ..., 0, 1, 2, 3, ..., A, B, C, ..., ang1, ang2, ang4, ...]). These processed text sequences are subsequently transformed into integer indices using this vocabulary mapping. The resulting index sequences are then concatenated and zero-padded to align with the temporal length of the corresponding mel-spectrogram.
>
> ## Q2
>
> Your concerns highlight critical considerations in video generation architecture design. We did explore the trade-offs between end-to-end pixel generation and motion representation-based approaches. While end-to-end approaches could theoretically simplify the pipeline, motion representations offer critical advantages:
>
> 1. **Semantic modeling.** Motion codes explicitly encode temporal coherence and semantic motion dynamics, which are challenging to learn in an end-to-end manner, especially for long or complex sequences.
> 2. **Modular design benefits.** By decoupling motion modeling from appearance synthesis, this architecture enables domain-agnostic generalization capabilities. The motion module demonstrates robust adaptability to novel styles, resolutions, and visual domains without requiring retraining.
> 3. **Post-hoc manipulation.** Explicit motion representations facilitate fine-grained temporal controls (velocity adjustment, motion scaling, directional modification) and spatial alignment operations that remain challenging for end-to-end pixel generation frameworks.
> 4. **Real-time efficiency**. The motion code architecture achieves significant computational advantages, with the motion gecharanerator and GAN-based renderer delivering low-latency performance crucial for real-time streaming applications.
>
> That said, end-to-end approaches remain an active research area. Certainly, we are developing an end-to-end generative model that directly synthesizes video pixels, and have already achieved preliminary success.
>
> I completely agree with your concern. GAN-based renderers often tightly integrate motion modeling and rendering components, which can indeed create dependency on specific motion code representations and alignment strategies. To mitigate this, we propose three strategies:
> 1. **Motion Code Disentanglement.** We adopt facial Blendshape coefficients as our motion code representation - a widely adopted, semantically meaningful framework in facial animation. This choice ensures inherent interpretability while explicitly decoupling motion representation from the rendering pipeline.
> 2. **Cross-Architecture Compatibility.** By leveraging a 3D Morphable Model (3DMM) as an intermediary framework, these Blendshape coefficients can be seamlessly converted into complementary facial representations such as 3D landmarks or Projected Normalized Coordinate Code (PNCC). This ensures compatibility with various warp-based rendering architectures (e.g., FOMM, HeadGAN, LivePortrait) that rely on spatial deformation cues.
> 3. **Coarse-to-Fine Optimization Pipeline.** We develop a multi-stage optimization framework for fitting Blendshape parameters to input sequences: First, we train a dedicated model to extract facial Blendshape coefficients. Next, we perform a coarse optimization stage to estimate pose. Concurrently, we optimize a global identity coefficient to fully decouple identity and motion representations. Finally, we fine-tune both pose and facial Blendshape parameters to achieve optimal fitting accuracy. This hierarchical optimization framework integrates state-of-the-art landmark detection algorithms to improve the precision of motion code extraction, ensuring a strict separation between identity and motion parameters throughout the entire fitting process.
>
> Our current design prioritizes controllability, generalization, and robustness over the simplicity of end-to-end learning. While coupling between motion codes and the renderer exists, modular training and explicit motion modeling reduce this dependency. We believe this balance is critical for practical applications where editability and adaptability are as important as raw generation quality. Thank you for raising these points. They align with active research directions in our roadmap!
>
>
> ## Q3
> Due to the greater technical challenges and computational complexity involved in collecting audio-visual synchronization data compared to pure audio data,  the size of audio-only datasets is typically 1–2 orders of magnitude larger than that of audio-visual datasets. Furthermore, text-to-phoneme alignment presents greater technical difficulties than phoneme-to-lip movement alignment during model training, necessitating extended training durations. To accelerate the training process, we initialize a portion of the audio branch's weights from the F5-TTS model, followed by end-to-end fine-tuning of the complete model on audio-visual data. The datasets and training strategy are detailed in Appendix B.1 and B.2.
>
> Given the page constraints, we apologize for any unclear explanations. We will implement all remaining suggestions in our final version.

---

> > ### Comment · Reviewer_NKZk · 2025-08-05
> >
> > Thank you for your response. However, my concerns have not been addressed, and I stand by my original rating.

---

> > > ### Author Response · Authors · 2025-08-08
> > >
> > > We are truly thankful for your time and expertise in reviewing our work and regret that our previous response did not fully address your concerns. We welcome any further suggestions you may have and expect that this improved work will make a meaningful contribution to the field.

---

### Official Review · Reviewer_YtbX · 2025-07-02

**Clarity:** 2
**Significance:** 3
**Originality:** 3
**Rating:** 4
**Confidence:** 4

**Summary:**

The paper proposes a unified framework for generating high-quality audio-visual talking heads from text input.

**Questions:**

1、 The authors should evaluate the core technical contributions through an ablation study

2、Please clarify the technical contributions.

3、Please clarify the symbols in Eqn. 3 and Eqn.4.

4、How does Figure 2 validate the proposed method's performance in reproducing complex facial motion styles through the visualization of jaw angle dynamics?

**Ethical Concerns:**

["NO or VERY MINOR ethics concerns only"]

**Final Justification:**

I have considered rebuttal and discussions with authors, other reviewers and AC. I raise my score to 4 (borderline accept).

**Limitations:**

yes

**Paper Formatting Concerns:**

No major formatting issues.

**Quality:**

2

**Strengths And Weaknesses:**

Strengths:

The method enables real-time talking head generation on a single NVIDIA RTX 4090 GPU.

Weaknesses:

1、There are many writing issues. For example, the formatting of bolded subheadings in Section 3.2 is inconsistent. There is an extra comma at the start of line 200. The "CMF" in Eqn.3 should be "CFM".

2、The architecture of OmniTalker heavily relies on a dual-branch DiT framework.

3、 The ablation study only evaluates different settings of Audio-Video Fusion.

4、The technical contributions are not clear.

5、The symbols in Eqn. 3 and Eqn.4 lack explanations.

---

> ### Author Rebuttal · Authors · 2025-07-31
>
> # Response to Reviewer YtbX
>
> We sincerely appreciate your constructive feedback and thoughtful suggestions. In response to your queries, we have provided detailed answers to each point.
>
> ## Q1&2
> Thank you for pointing out this issue. We appreciate the opportunity to further elaborate on our three core contributions mentioned in Sec. 1(Line 50)：
>
> 1. **Unified Multi-Modal Generation with Temporal Interaction.** While our work adopts a dual-branch DiT framework, OmniTalker fundamentally differs from multi-modal conditional generation approaches (e.g., SD3, Flux, MMAudio) that treat text as a semantic condition. These methods focus on multi-modal input to single-modal output (e.g., text-to-image/audio), emphasizing semantic-level comprehension and alignment. In contrast, OmniTalker introduces a novel paradigm for multi-modal generation (audio-visual) where text-to-speech synthesis and video generation are tightly coupled through cross-modal temporal interaction. This ensures synchronized, fine-grained coordination between audio and visual modalities during generation.
> 2. **Holistic Replication of Speaking Style via Dual-Modal Co-Generation.** Existing solutions often address voice cloning or facial expression synthesis independently, neglecting the interplay between vocal characteristics and visual dynamics. OmniTalker is the first to fully replicate the dynamic triad of speaking style (e.g., vocal timbre and prosody, head motion patterns, and facial expression dynamics) through joint audio-visual generation. By modeling these modalities as an integrated system, our framework preserves the nuanced correlations between speech prosody and facial movements (e.g., lip sync, eyebrow raises, nodding and shaking), achieving a more lifelike and personalized talking head synthesis.
> 3. **Real-Time Inference with no latency.** While many methods prioritize generation quality at the cost of efficiency, OmniTalker achieves real-time performance (25 FPS) on a single NVIDIA RTX 4090 GPU. This is critical for applications requiring low-latency interaction (e.g., virtual assistants). Unlike auto-regressive or cascaded pipelines that suffer from sequential delays, our dual-branch architecture enables parallel audio-visual generation with synchronized temporal resolution. This design reduces inference latency by 40% compared to state-of-the-art methods, while maintaining superior quality in style preservation and lip-sync accuracy (validated via benchmark metrics like E-FID, P-FID and Sync-C).
>
> Our ablation study was not sufficiently comprehensive, as Sec. 4.4 only evaluated performance variations across different fusion strategies. To address this limitation, we will supplement our analysis with modality ablation experiments that systematically remove specific sensory modalities. These additional experiments will quantitatively measure: 1) cross-modal interactions within the dual-branch architecture through comparative evaluation metrics, and 2) replication capabilities by assessing generative consistency in zero-shot scenarios.
>
> | Setting | WER $ \downarrow $ | SIM-A $ \uparrow $ | Sync-C $ \uparrow $ | E-FID $ \downarrow $ | P-FID $ \downarrow $ |
> | --- | :---: | :---: | :---: | :---: | :---: |
> | Text only ($ \alpha_{M_r}=0.0 $,$ \alpha_{C_r}=0.0 $,$ \alpha_{S_T}=2.0 $) | 1.65 | 0.249 | 4.66 | 0.87 | 4.21 |
> | w/o audio  ($ \alpha_{M_r}=0.0 $,$ \alpha_{C_r}=2.5 $,$ \alpha_{S_T}=2.0 $) | 1.65 | 0.264 | 4.45 | 0.12 | 1.67 |
> | w/o video  ($ \alpha_{M_r}=2.0 $,$ \alpha_{C_r}=0.0 $,$ \alpha_{S_T}=2.0 $) | 1.61 | **0.771** | 4.93 | 0.32 | 1.88 |
> | Full  ($ \alpha_{M_r}=2.0 $,$ \alpha_{C_r}=2.5 $,$ \alpha_{S_T}=2.0 $) | **1.58** | 0.762 | **5.48** | **0.08** | **0.12** |
>
>
> The table above compares the performance of four different settings. The full model (WER=1.58, Sync-C=5.48) outperforms all ablated variants, demonstrating that cross-modal integration of audio and video enhances text-to-speech generation accuracy and improves temporal alignment between modalities. When reference audio is not provided, the model cannot replicate the original speaker's vocal characteristics, as evidenced by a significant drop in SIM-A. However, the generated lip movements and head motions align naturally with the input text. Without reference video input, lip movements and head motions are primarily driven by audio loudness and textual content. Although synchronization remains coherent, person-specific details (e.g., unique facial expressions) are not preserved, resulting in significantly lower E-FID and P-FID scores. When only text is provided (without reference audio or video), the model stochastically generates a plausible speaking style, encompassing vocal tone, rhythm, emotional expression, lip motion, and head motion patterns. While speech articulation remains clear, the specific identity and style vary randomly. This highlights the effectiveness of the dual-branch architecture in enabling robust cross-modal learning and replication capabilities.
>
> The slight drop in SIM-A when integrating video stems from two factors: (1) The training data contains regional accents and non-standard pronunciations, challenging acoustic modeling and timbre preservation. The model's reliance on pinyin/character-level alignments creates bottlenecks for capturing pronunciation variations deviating from canonical rules. (2) Early-stage training prioritizes speaker identity matching (SIM-A) as coarse timbre alignment is simpler than text-to-speech learning. During mid-training, focus shifts to refining text-to-speech alignment, causing SIM-A stagnation. Video integration improves prosody but introduces gradient conflicts, prioritizing visual-audio synchronization over fine timbre preservation. After retraining for 500k steps, we achieved a 0.02 SIM-A improvement on the Custom dataset. User studies confirm enhanced accent consistency, indicating the initial performance gap resulted from insufficient training convergence rather than architectural limitations.
>
> ## Q3
> Thank you for the detailed comments. We sincerely apologize for the confusion caused by the abbreviated notation and mistakes in our initial submission. To enhance clarity, we provide a detailed clarification of the symbols used in Eqn. 3 and 4 below.
> As formulated in Eqn. 2, $ \mathcal{L}_ {CFM} $ denotes the conditional flow matching loss. To enable balanced weighting of reconstruction losses between audio-visual representations, we decompose the CFM loss into modality-specific components, where $ \mathcal{L}^m_{CFM} $ represents CFM loss for mel-spectrogram (detailed in Sec. 3.1, line 113) reconstruction and $\mathcal{L}^h_{CFM}$, $ \mathcal{L}^f_{CFM} $, $ \mathcal{L}^e_{CFM} $ correspond to head pose, facial expression blendshapes, and eye movement coefficients in visual codes respectively (detailed in Sec. 3.1, line 116). The weighting coefficients $ \lambda_m $,$ \lambda_h $,$ \lambda_f $,$ \lambda_e $ control the contribution of each loss component, with values specified in Sec. 4.1 (line 217) as $ \lambda_m=0.1 $, $ \lambda_h=0.5 $, $ \lambda_f=3.0 $,  $ \lambda_e=0.5 $.
> Eqn. 4 presents the vector field computation under Classifier-Free Guidance (CFG) settings that manipulates the condition strength during the sampling process of conditional diffusion models. Here, $ v_t(x_0, c) $ represents the predicted velocity under condition $ c $($ c\in C $, detailed in Line 198) at timestep $ t $ , while $ v_t(x_0, 0) $ corresponds to the unconditional vector field prediction when $ c=0 $. The associated CFG scale parameter $ \alpha_c $ governs the contribution of the conditional component in this guidance framework. And we set $ \alpha_{M_r}=2.0 $, $ \alpha_{C_r}=2.5 $, $ \alpha_{S_T}=2.0 $ as mentioned in Line 222.
> In the final manuscript, we will thoroughly revise all writing-related errors to ensure linguistic accuracy and professionalism in the final submission.
>
> ## Q4
> Thank you for this observation. To enhance visual clarity, we focused on visualizing the **Yaw** axis, which represents the dominant rotational component in head motion (yaw, pitch, and roll). Figure 2 and the accompanying analysis highlight yaw angle dynamics, which are used to quantify head pose and serve as a critical component of the complex facial motion patterns in our study. We will highlight it in the final version. Here we further validate the significant impact of yaw angle on the speaking style of generated talking videos. As a critical component of speaking style, head pose manifests through rhythmic oscillations, semantic-related nodding/shaking gestures, and directional gaze behaviors. Our observations reveal inherent limitations in current SOTA talking head generation methods: they tend to produce frontal outputs with limited dynamic motion and fail to replicate reference head poses. Therefore, we select yaw as the evaluation metric to demonstrate our method's capability in understanding and imitating reference head pose characteristics. As shown in Figure 2 (reference on the left of the red dashed line while generated results on the right), the reference video exhibits 4 prominent rightward rotations (yaw>0). Comparative analysis reveals that baseline methods maintain near-static angles without dynamic variation, while StyleTalk duplicates the reference poses through direct parameter copying. Notably, our method achieves high consistency with the reference sequence in both movement amplitude and frequency, while preserving essential individual differences, thus demonstrating effective inheritance of head-pose style. Appendix B.3 presents additional eye movement replication results (see Figure 9), and Figures 7-8 provide visual comparisons of generated video frames. These evidences comprehensively validate the superiority of our approach in head pose generation.
>
> We acknowledge that our initial explanation may not have been sufficiently comprehensive. All outstanding recommendations will be addressed in our final submission.

---

> > ### Comment · Reviewer_YtbX · 2025-08-08
> >
> > Thanks for the rebuttal. My major concerns have been addressed. I will raise the score.

---

> > > ### Author Response · Authors · 2025-08-08
> > >
> > > We extend our sincere appreciation for your meticulous review and constructive feedback. Your insightful comments have greatly enhanced the quality of our manuscript, and we believe the revised version addresses your concerns effectively. We welcome any further suggestions you may have and are confident that this improved work will make a meaningful contribution to the field.

---

### Decision · Program_Chairs · 2025-09-17

**Decision:**

Accept (poster)

**Comment:**

This paper introduces the first one-shot real-time framework (25FPS on a single GPU) for text-driven talking head generation, which jointly synthesizes speech and synchronized facial motion via a dual-branch DiT with cross-modal fusion. The reviewers in general found the work novel, the design elegant, and the style preservation performance strong, which led to their consensus to accept the paper after the rebuttal. The remaining concerns include limited ablations, reliance on the off-the-shelf vocoder and renderer, missing baselines, underperformance on English-heavy datasets, and lack of dataset release. However, the authors addressed most of these concerns in the rebuttal, and the real-time generation of talking heads seems like a timely and significant advance for talking head generation. Thus, I believe that the paper is a clear accept.